# Eco-alternative treatments for *Vibrio parahaemolyticus* and *V. cholerae* biofilms from shrimp industry through Eucalyptus (*Eucalyptus globulus*) and Guava (*Psidium guajava*) extracts: A road for an Ecuadorian sustainable economy

**Nicolás Renato Jara-Medina[1], Dario Fernando Cueva[1], Ariana Cecibel Cedeño-Pinargote[1], Arleth Gualle[2], Daniel Aguilera-Pesantes[3], Miguel Ángel Méndez[2], Lourdes Orejuela-Escobar[2,4,5], Diego F. Cisneros-Heredia[6], Rebeca Cortez-Zambrano[1,6,7], Nelson Miranda-Moyano[6], Eduardo Tejera[8]\*, António Machado[1]\***

**1** Laboratorio de Bacteriología, Colegio de Ciencias Biológicas y Ambientales COCIBA, Instituto de Microbiología, Calle Diego de Robles y Pampite, Universidad San Francisco de Quito USFQ, Quito, Ecuador, **2** Colegio de Ciencias e Ingenierías, Departamento de Ingeniería Química, Universidad San Francisco de Quito USFQ, Quito, Ecuador, **3** Research, Development, and Strategic Planning Department, Clydent S.A., Guayaquil, Ecuador, **4** Instituto de Investigaciones Biológicas y Ambientales BIÓSFERA, Universidad San Francisco de Quito USFQ, Quito, Ecuador, **5** Instituto de Investigaciones en Biomedicina, Universidad San Francisco de Quito USFQ, Quito, Ecuador, **6** Colegio de Ciencias Biológicas y Ambientales, Instituto de Biodiversidad Tropical IBIOTROP, Herbario de Botánica Económica, Universidad San Francisco de Quito USFQ, Quito, Ecuador, **7** Facultad de Medicina Veterinaria, Universidad UTE, Quito, Ecuador, **8** Facultad de Ingeniería y Ciencias Agropecuarias Aplicadas, Grupo de Bioquimioinformática, Universidad de Las Américas (UDLA), Quito, Ecuador

\* amachado@usfq.edu.ec (AM); eduardo.tejera@udla.edu.ec (ET)

## Abstract

Understanding how environmental variables influence biofilm formation becomes relevant for managing *Vibrio* biofilm-related infections in shrimp production. Therefore, we evaluated the impact of temperature, time, and initial inoculum in the biofilm development of these two *Vibrio* species using a multifactorial experimental design. Planktonic growth inhibition and inhibition/eradication of *Vibrio* biofilms, more exactly *V. parahaemolyticus* (VP87 and VP275) and *V. cholerae* (VC112) isolated from shrimp farms were evaluated by Eucalyptus and Guava aqueous leaf extracts and compared to tetracycline and ceftriaxone. Preliminary results showed that the best growth conditions of biofilm development for *V. parahaemolyticus* were 24 h and 24°C ($p$ <0.001), while *V. cholerae* biofilms were 72 h and 30°C ($p$ <0.001). Multivariate linear regression ANOVA was applied using colony-forming unit (CFU) counting assays as a reference, and R-squared values were applied as goodness-of-fit measurements for biofilm analysis. Then, both plant extracts were analyzed with HPLC using double online detection by diode array detector (DAD) and mass spectrometry (MS) for the evaluation of their chemical composition, where the main identified compounds for Eucalyptus extract were cypellogin A, cypellogin B, and cypellocarpin C, while guavinoside A, B, and C compounds were the main compounds for Guava extract. For planktonic growth

**Data Availability Statement:** All data may be found in the paper and its Supporting Information files.

**Funding:** This work is supported by the COCIBA research budget to António Machado from Universidad San Francisco de Quito, under the Project ID: 17357 entitled "Alternative approaches for eliminating Biofilms" and the Project ID: 16801 entitled "Characterization of single and mixed Biofilms". The funders had no role in the study design, data collection, analysis, decision to publish, or preparation of the manuscript. The APC was funded by the Research Office of Universidad San Francisco de Quito (USFQ).

**Competing interests:** The authors have declared that no competing interests exist.

inhibition, Eucalyptus extract showed its maximum effect at 200 µg/mL with an inhibition of 75% ($p < 0.0001$) against all *Vibrio* strains, while Guava extract exhibited its maximum inhibition at 1600 µg/mL with an inhibition of 70% ($p < 0.0001$). Both biofilm inhibition and eradication assays were performed by the two conditions (24 h at 24°C and 72 h at 30°C) on *Vibrio* strains according to desirability analysis. Regarding 24 h at 24°C, differences were observed in the CFU counting between antibiotics and plant extracts, where both plant extracts demonstrated a higher reduction of viable cells when compared with both antibiotics at 8x, 16x, and 32x MIC values (Eucalyptus extract: 1600, 3200, and 6400 µg/mL; while Guava extract: 12800, 25600, and 52000 µg/mL). Concerning 72 h at 30°C, results showed a less notorious biomass inhibition by Guava leaf extract and tetracycline. However, Eucalyptus extract significantly reduced the total number of viable cells within *Vibrio* biofilms from 2x to 32x MIC values (400–6400 µg/mL) when compared to the same MIC values of ceftriaxone (5–80 µg/mL), which was not able to reduce viable cells. Eucalyptus extract demonstrated similar results at both growth conditions, showing an average inhibition of approximately 80% at 400 µg/mL concentration for all *Vibrio* isolates ($p < 0.0001$). Moreover, eradication biofilm assays demonstrated significant eradication against all *Vibrio* strains at both growth conditions, but biofilm eradication values were substantially lower. Both extract plants demonstrated a higher reduction of viable cells when compared with both antibiotics at 8x, 16x, and 32x MIC values at both growth sets, where Eucalyptus extract at 800 µg/mL reduced 70% of biomass and 90% of viable cells for all *Vibrio* strains ($p < 0.0001$). Overall results suggested a viable alternative against vibriosis in the shrimp industry in Ecuador.

# 1 Introduction

The aquaculture industry according to the Food and Agricultural Organization (FAO) is one of the industries with the highest production growth with 9.4% per year [1]. However, it has been affected by aquatic diseases mainly produced by various species of *Vibrio* [2], which generates economic losses for all crustacean-producing countries. According to the Global Outlook for Aquaculture Leadership (GOAL) of 2019, Ecuador is the country with the largest production and export of shrimp in the world, and, in the last decade, it has had a rapid growth of approximately 16% [3]. Additionally, it represents 19% of the country's exports, generating 5 billion dollars in 2021. The shrimp industry is located in the coastal zone of the country, distributed mainly in 5 provinces, with Guayas being the province with the highest production (60%), followed by El Oro, Esmeraldas, Manabí, and Santa Elena [4]. However, one of the main drawbacks of the shrimp aquaculture industry is *Vibrio*-related infections [5]. Probiotics are usually chosen as the best alternatives to these antimicrobial agents, and they act as natural immune enhancers, which provoke disease resistance in shrimp aquaculture [6, 7]. However, green chemistry constitutes nowadays a pathway for a sustainable economy and future to the efficient and clean transformation of renewable raw materials into functional chemicals [8]. Eucalyptus and Guava leaves constitute raw and cheap materials with well-known antimicrobial activity [9, 10], which are not commonly used in the aquaculture industry, thus constituting a source of antimicrobial agents for a new potential sustainable economy for the Ecuadorian market.

*Vibrio* spp. is a curved rod gram-negative bacterium, facultative aerobic and facultative anaerobic rods, which has a single flagellum [11]. It is generally found in warm salty aquatic

environments such as seawater and estuaries. This genus is also found in different conditions, survives at 5–45°C, and achieves substantial growth when seawater temperatures are over 14–19°C [12, 13]. It is an opportunistic pathogenic bacterium for marine animals as well as for people [14]. In aquaculture, this pathogen can be a big issue causing a deadly disease for shrimp and an opportunistic gastrointestinal infection in humans [15]. In nature, *Vibrio* species can usually be found in multispecies communities that form biofilms, which gives them greater resilience and adaptability [16]. Biofilms are microbial consortia that are enveloped and protected by an extracellular polymeric substance (EPS) matrix, being the most common bacterial community found in nature. Biofilms are formed due to quorum sensing (QS), which is a communication system used by bacteria that produces changes in gene expression driven by a message (c-di-GMP) [17]. This causes a phenotypic shift with the reduction of the motor activity of the flagella and regulates the expression of surface proteins for the union and synthesis of the extracellular matrix [18]. Although all *Vibrio* species are ubiquitous in aquatic ecosystems and can establish strong biofilms, *Vibrio* species can also differ in the mechanisms and regulation of their biofilm formation [19]. For example, *V. cholerae* biofilms are positively controlled by VpsR and VpsT regulators via the transcriptional control of *vps* genes, where VpsR is greater than VpsT. Meanwhile, *V. parahaemolyticus* biofilms are positively regulated via expression of *cps* genes, which are negatively regulated by a homolog of VpsT known as CpsS. In the absence of CpsS and CpsR (VpsR homolog), the positive *cps* gene expression leads to the *V. parahaemolyticus* biofilm formation. Thus, *V. cholerae* and *V. parahaemolyticus* use similar proteins, but they function in the opposite sense and to different degrees: CpsS is the dominant negative regulator in *V. parahaemolyticus* whereas VpsT is a positive co-regulator in *V. cholerae* [13, 19].

The main diseases that affect shrimp are the early mortality shrimp-acute hepatopancreatic necrosis disease (EMS–AHPND), which affects shrimp postlarvae within 20–30 days after stocking and frequently causes up to 100% mortality [20, 21], and vibriosis which mortality can range between 70–100% but damages the shrimp´s oral cavity and appendages [5, 22]. Therefore, several *Vibrio* species can significantly reduce shrimp production where *V. parahaemolyticus*, *V. harveyi*, *V. alginolyticus*, and *V. vulnificus* are commonly isolated in these shrimp farms [23]. In addition, certain *Vibrio* species can emerge as a serious threat to human health, such as *V. parahaemolyticus*, *V. cholerae*, and *V. alginolyticus* [24]. In Ecuador, the prevalence of *V. parahaemolyticus* and *V. cholerae* in shrimps at retail, farms, and estuaries is alarming and represents an eminent and serious public health risk [25, 26].

In marine biotic or abiotic surfaces under appropriate growth conditions, *Vibrio* sp. biofilms can function as a source of pathogenic bacteria with 10–1000 times more resistance against hygiene treatment than planktonic counterparts. In a study reported by Odeyemi (2016), the authors evaluated the prevalence of *V. parahaemolyticus* based on 48 published studies and reported a 48.3% of its presence in shrimp, especially in uneven seafood surface areas where pits and edges protected biofilm communities. According to international standards, there are certain antibiotics such as chloramphenicol, tetracyclines, sulphonamides, and quinolones, which cannot be used due to the presence of antibiotic residues in the shrimp that could be ingested by the consumers and therefore they are not allowed in the shrimp production [11, 27]. Some alternative strategies to control *Vibrio* species and their biofilm formation is using probiotics and natural products in mature ecosystems [2, 6]. In 2018, Yatip et al. discovered that extract from the fermented soybean product Natto (generally regarded as safe, GRAS) can inhibit *Vibrio* biofilm formation significantly reducing shrimp mortality with *V. harveyi* infection [28]. Likewise, Karnjana et al. reported that the ethanolic extract of the red seaweed *Gracilaria fisheri* enhanced immune activities and inhibited the growth of *V. harveyi* in shrimp suggesting a potential application of this *G. fisheri* extract as an efficient approach

for treating biofilm-associated *Vibrio* diseases in aquacultures [29]. Furthermore, Kamble et al. explored sulfated galactans degraded fractions of the *G. fisheri* evidencing a stronger antibacterial activity against *V. parahaemolyticus* and *V. harveyi* through the disruption of the bacterial cell membrane resulting in leakage of intracellular biological components and thus cell death [30]. In addition to preventing the growth and spread of *Vibrio* species due to competition, certain probiotic bacteria are also able to perform quorum quenching, which is the QS inhibition thus inhibiting biofilm formation [7]. However, when biofilms have already been established, these strategies don´t work well. Therefore, it is important to analyze the biofilm life cycle of the main pathogenic *Vibrio* species identifying the best growth conditions to further evaluate suitable treatments against *Vibrio*-related biofilms.

So, the present work aimed to evaluate the biofilm formation of *V. parahaemolyticus* and *V. cholerae* and characterize their biofilm production through different *in vitro* conditions. This study characterized *Vibrio* sp. biofilms at different temperature conditions (24 and 30˚C), during a range of time (24, 48, and 72 h), and at different concentrations of initial inoculum (1E+7 and 1E+8 colony-forming units (CFU)/mL); biomass growth assays (optical density measurement at 630 nm using crystal violet staining and PBS suspension); CFU counting assay; and, fluorescence microscopy (FM) analysis. After the establishment of the best experimental conditions for *Vibrio*-related biofilms, Guava (*Psidium guajava* L.) and Eucalyptus (*Eucalyptus globulus* Labill.) aqueous leaf extracts were further evaluated as eco-alternative treatments for *V. parahaemolyticus* and *V. cholerae* biofilms through biofilm inhibition and eradication assays and compared to standard antibiotic treatments, more exactly tetracycline and ceftriaxone.

## 2 Materials and methods

### 2.1 Bacterial isolates and growth conditions

*V. parahaemolyticus* and *V. cholerae* strains from the microbial collection of the Institute of Microbiology at Universidad San Francisco de Quito USFQ (IM-USFQ), designated as IMUSFQ-VP87, tetracycline-resistant IMUSFQ-VP275, and IMUSFQ-VC112, respectively, were selected for the present study. These *Vibrio* isolates were previously recovered by Clydent S.A. laboratory (https://clydent.com/) from diseased shrimp samples of shrimp farms located in El Oro, Santa Elena, and Guayas provinces, and then identified through DNA sequences at multiple loci and biochemical properties at the Clydent S.A. laboratory and IM-USFQ. These isolates were stored at -80˚C or -20˚C, and 24 h before each assay, a new culture in trypticase soy agar (TSA; Dipco Cía. Ltda., Quito, Ecuador) supplemented with NaCl 1% (w/v; see Supplemental Material) was made to avoid natural mutants [31, 32]. After 24 h of growth culture at 30˚C, bacterial cells were harvested and suspended in phosphate-buffered saline (PBS) to obtain a cellular density equal to 1E+8 CFU per mL using 0.5 McFarland turbidity standard and then a second dilution 1:10 was prepared to obtain another concentration of 1E+7 CFU/ mL gathering two initial inoculum concentrations for biofilm assays. The optical density of both *Vibrio* inoculums was measured at 630 nm using a UV-Vis spectrometer (GENESYS™ 20 Thermo Scientific™, Waltham, Massachusetts, USA), ensuring the identical concentration in all assays (see Supplemental Material).

### 2.2 Plant extract and chemical analysis

Fresh guava (*Psidium guajava*) leaf samples were obtained from the Sangolquí market, province of Pichincha, Ecuador. Eucalyptus (*Eucalyptus globulus*) leaf samples were collected at Alangasí, province of Pichincha, Ecuador. All samples were identified by Nelson Miranda, curator at the Herbario de Botánica Económica del Ecuador of the Universidad San Francisco

de Quito USFQ. The raw materials were washed and dried in an oven at 40°C for 24 h. The dry leaves were then ground and sieved to a particle size of less than 250 microns. Soxhlet extractions at 80°C were carried out with 10 grams of dry biomass and 300 mL of distilled water for 3 h. Percent yield values of both plant extracts were calculated using the following formula:

$$Yield\ (\%) = \frac{Dried\ extract\ weight\ (g)}{Biomass\ weight\ (g)} \times 100.$$

For every plant extract, triplicate leaf samples were used for extraction and yield values of the extracts were expressed as mean percentage ± standard deviation (SD) (n = 3).

The dry extracts were dissolved in 1 mL of methanol/water (80:20, v/v), filtered through a 0.22 μm disposable LC filter disk (RephiLe Bioscience Ltd), and analyzed with HPLC using double online detection by diode array detector (DAD) and mass spectrometry (MS) following a previously described methodology [33]. The system consisted of a Vanquish (Thermo Fisher Scientific) fitted with a binary pump and DAD coupled with an LTQ-XL (Thermo Fisher Scientific) controlled by Xcalibur Software (Thermo Scientific™, Waltham, Massachusetts, USA). An Accucore Vanquish C18 column (1.5 μm, 100 × 2.1 mm) thermostated at 35°C was used as the stationary phase, while the mobile phase consisted of a solution of 0.1% formic acid (A) and acetonitrile (B). The elution gradient established was: 2% B, 0–4 min; 4% B, 4–22 min; 40% B, 22–32 min; 70% B, 32–35 min; 98% B, 35–40 min, and rebalanced the column to the initial conditions of the solvent. The flow rate was 0.2 mL/min, and the injection volume was 50 μL (see the descriptive chromatograms in S1 Fig). Double line detection was carried out in DAD at 280, 330, and 370 nm and MS operated in positive ion mode. The dependent data analysis (DDA) was done on the 5 most intense ions with a normalized collision energy of 35. Spectra between m/z 100 and m/z 1500 were recorded. The Electrospray Ionization (ESI) conditions were as follows: capillary temperature of 275°C, source voltage and capillary voltage of 5Kv and -35 V, respectively, and tubelens –200. The peak tentative identification was exclusively done using MzCloud (https://www.mzcloud.org/) and Global Natural Products Social molecular networking (GNPS, http://gnps.ucsd.edu) databases as well as the scientific literature.

### 2.3 Ethics statement

Voucher specimens of Guava (*Psidium guajava*) and Eucalyptus (*Eucalyptus globulus*) leaf samples used in this study are deposited at Herbario de Botánica Económica del Ecuador of the USFQ under the codes #35004 and #35005, respectively. The present study was conducted under the authorization for collection of biological specimens (MAATE-ARSFC-2024-0328) issued by the Ministry of Environment, Water and Ecological Transition of Ecuador MAATE.

### 2.4 Biofilm assays

The two previous inoculums were evaluated in the present assays of *Vibrio*-related biofilm formation. Each inoculum of *V. cholerae* or *V. parahaemolyticus* was centrifuged at 10000 rpm, for 20 min and the pellet was resuspended in sterile trypticase soy broth (TSB; Dipco Cía. Ltda., Quito, Ecuador) supplemented with NaCl 1% (w/v). In each well of the 6-well plate containing a sterile coverslip, 3 mL of primary biofilm inoculum was added. Also, blank or sterility control was prepared on the same plate, which only contained a cover slip in a sterile medium [31]. Plates containing different inoculums and *Vibrio* species were also incubated at two different temperatures (24 and 30°C) and for different periods (24, 48, and 72 h) under static conditions, replacing the medium in each well with 3 mL of fresh medium every 24 h after the

biofilm samples were washed with 3 mL of PBS [34]. Each condition setting (*Vibrio* species, initial inoculum, temperature, and growth culture period) was performed with at least six assays, and, in each assay, five biofilm samples were separately prepared. All biofilm assays were performed on different days.

## 2.5 Biomass quantification

To screen the ability of each strain of *V. cholerae* and *V. parahaemolyticus* to form a biofilm, we evaluated biomass formation using an optical density (OD) assay with crystal violet (CV) staining and phosphate-buffered saline (PBS) suspension, as prepared in our previous study [31]. Briefly, each optical density assay is described below.

**2.5.1 Crystal violet staining.** After a certain period of growth (24, 48, and 72 h), we used the CV method to quantify the biomass of biofilm samples as described by Peeters and colleagues [35] with slight modifications. Briefly, the biofilm samples on coverslips were carefully washed four times by submerging each coverslip in a well with 3 mL of sterile PBS. Then, the coverslips containing the biofilm sample were transferred to a clean 6-well plate and stained with 3 mL of crystal violet 1% (v/v) for 30 min, and the excess staining was carefully removed from the wells. 3 mL of alcohol 96% (v/v) were placed into each well for 5 min for fixation and then each coverslip containing the fixed biofilm sample was transferred into a sterile plastic flask with 3 mL of alcohol 96% (v/v), and vortexed at maximum velocity for 15 min to ensure the biofilm removal of the coverslip into the alcohol solution. Finally, 200 µL of each biofilm sample was placed in a 96-well plate and read in the ELISA Elx808 spectrophotometer (BioTek, Winooski, USA) at 630 nm. All biofilm samples and sterility controls (coverslips in medium with no inoculum) were included in the OD measurements on the 96-well plate. All OD measurements by CV staining were adjusted by subtracting the absorbance measurements of sterility controls from the absorbance measurements of biofilm samples.

**2.5.2 Phosphate-buffered saline suspension.** Likewise, the second set of biofilm samples was carefully washed four times with 3 mL of sterile PBS. Then each coverslip containing the biofilm sample was transferred in a sterile plastic flask with 3 mL of sterile PBS, and vortexed at maximum velocity for 15 min to ensure the biofilm removal of the coverslip into the PBS solution. For each sample, 200 µL of the previous suspension were added to a 96-well plate and read in the ELISA Elx808 spectrophotometer at 630 nm. As previously mentioned, all biofilm samples and sterility controls (coverslips in medium with no inoculum) were measured. All OD measurements by PBS suspension were adjusted by subtracting the absorbance measurements of sterility controls from the absorbance measurements of biofilm samples. The remaining PBS suspension was used for viability quantification assays.

## 2.6 Viability quantification

**2.6.1 Colony-forming units counting assays.** To enumerate culturable sessile cells, CFU counting assays were used. At least three individual PBS suspensions of each biofilm sample (resuspended *Vibrio* sp. biofilm of the initial coverslip) were used in a serial tenfold dilution, by adding 100 µL of sample to 900 µL of sterile PBS supplemented with NaCl 1% (w/v). Each dilution was thoroughly vortexed and pipette tips were changed before the next dilution or experimental step. The tested dilutions included (1E-5 –1E-7) and each dilution was plated on TSA supplemented with NaCl 1% (w/v) by triplicate results per biofilm sample. All plates were incubated for 24 h at 30˚C, after which colonies were counted. The experiments were performed at the same time as biomass experiments; thus, three CFU assays per dilution were available for analysis, and data was collected. For statistical analysis, the dilution with a growth between 25 and 150 CFU was chosen according to our previous study [31].

**2.6.2 LIVE/DEAD assays.** After the evaluation of CFU counting and biomass quantification assays, all time samples (24, 48, and 72 h) were chosen to be analyzed by fluorescence staining. After each biofilm formation assay in 6-well plates, any remaining medium in the wells was removed and coverslips were transferred to a new and sterile 6-well plate. A working solution of fluorescent stains was prepared by adding 1.0 mL of SYTO® 9 stain and 10 μL of propidium iodide (PI) stain (FilmTracer™ LIVE/DEAD® Biofilm Viability Kit, Invitrogen, Carlsbad, California, USA), mixed in the proportion 1:100 of PI/SYTO-9, into 10 mL of filter-sterilized water in a foil-covered container. This live/dead working solution was stored at -20˚C. 200 μL of the live/dead working solution were added onto each coverslip (biofilm sample), gently so it would not disturb the biofilm. All samples were then incubated for 15–30 min at room temperature, protected from light, before being rinsed with 200 μL of PBS. Finally, after the water evaporation of the coverslips, each coverslip was then placed face up onto a clean, dry microscope slide and a drop of mounting medium was added (ProLong Gold Anti-fade, ThermoFisher Scientific, Massachusetts, USA). An autoclavable 22-mm diameter glass coverslip (Dipco Cía. Ltda., Quito, Ecuador) was used to fix the sample in place. Samples were stored protected from light at room temperature (25 ˚C) until epifluorescence microscopy analysis, which was performed within the first 3 h [31, 36, 37].

**2.6.3 Fluorescence microscopy.** Fluorescence microscopy (FM) was carried out using an Olympus BX50 microscope (Olympus Corporation, Tokyo, Japan) equipped with a 100x oil immersion objective. Images were captured with AmScope Digital Camera MU633-FL (AmScope, California, USA) and digitalized with AmScope software version 1.2.2.10. As previously described by Cabezas-Mera and colleagues [34], for counting purposes at least 12 images were taken per sample on the 22-mm diameter glass coverslip at random locations. In addition, both *V. parahaemolyticus* and *V. cholerae* exhibited mature biofilms with multilayered growth over time, due to this problem the biofilm sample was resuspended in 10 mL of PBS and vortexed at maximum velocity for 5 min. Then, a 30 μL aliquot was placed on a coverslip, thus facilitating the cell area count in each biofilm and comparing it between *Vibrio* species under different growth conditions, as previously reported [38]. The total cell counts were given per cm$^2$ ± standard deviation ($N$ cells/cm$^2$ ± SD) by manually counting the number of *Vibrio* cells from each field (12,880 μm$^2$) and recalculating the average number of cells over the total coverslip area (4.84E+8 μm$^2$), as previously described [39]. In FM analysis, the percentages of dead and alive cells within images were measured through ImageJ version 1.57 by Fiji [40] using the macros Biofilms Viability checker proposed by Mountcastle and colleagues [37] and the plugin MorphoLibJ [41]; while, the total cells counting images were processed by a sequence of modules forming a pipeline designed for this purpose in Cell Profiler software [42] an open-source software version 4.2.1 (available from the Broad Institute at www. cellprofiler.org), the applied pipeline of which can be revised in Supplemental Information.

## 2.7 Antimicrobial activity

After the optimization and evaluation of biofilm growth conditions, the antimicrobial activity of the plant extracts (Eucalyptus and Guava leaves) was evaluated and compared with well-known antibiotics (tetracycline and ceftriaxone) against *Vibrio* pathogens through the microdilution method for minimum inhibitory concentration (MIC) assays and biofilm inhibition and eradication assays. The antimicrobial activity was evaluated by biomass quantification and CFU counting assays.

**2.7.1 Minimum inhibitory concentration.** The microdilution method for MIC assays was conducted according to Wiegand and colleagues [43], under the Clinical Laboratory Standards Institute (CLSI) guidelines. In MIC assays, serial dilution was prepared to start with 80 μg/mL antibiotics (tetracycline and ceftriaxone) and 1600 μg/mL plant extracts (Eucalyptus

and Guava leaves) powder in 190 μl of TSB (Dipco Cía. Ltda., Quito, Ecuador) supplemented with NaCl 1% (w/v) in 96-well plates [44]. Then, the remaining wells of the plate were set up with the diluted concentrations of 40, 20, 10, 5, and 2 μg/mL of tetracycline and ceftriaxone, while 800, 400, 200, 100, and 50 μg/mL of Eucalyptus and Guava extracts within the total volume of 200 μL of TSB broth with 1%(w/v) NaCl in the 96-well plate. All wells were prepared with a final concentration of 1E+5 CFU/mL of *Vibrio* inoculum, excepting negative controls, and the 96-well plate was incubated for 18–24 h at 37˚C [43]. A row of TSB with 1% (w/v) NaCl plus pathogen inoculum was used as culture growth control (or positive control), and a row of wells filled with medium without inoculum was applied to sterility control (negative control). Tetracycline and ceftriaxone (Sigma-Aldrich, St. Louis, MO, USA) were used as a reference control [33]. The MIC values were first evaluated by the unaided eye and then measured with the spectrophotometer Biotek Instruments ELx808IU at 630 nm (OD630) by comparing the results obtained with the negative control. The MIC was defined as the lowest concentration of antibiotic or plant extract that inhibited the *Vibrio* growth after overnight incubation. All the assays were independently performed in triplicate on different days.

**2.7.2 Biofilm inhibition and eradication assays.** An overnight culture of the *Vibrio* species was grown and an inoculum of 1E+7 CFU/mL in sterile TSB supplemented with NaCl 1% (w/v) was prepared as previously described and 190 μL were added to the 96-well plate. Next, the wells were supplemented with 10 μL of concentrated 20x antibiotics (tetracycline and ceftriaxone) or plant extracts (Eucalyptus and Guava) achieving 200 μL of the final volume in each well setting up 1x to 32x MIC values [45]. Plates were then incubated under two different conditions at 24 h at 24˚C and 72 h at 30˚C under aerobic conditions to evaluate the biofilm inhibition. Meanwhile, in biofilm eradication assays, 200 μL of media plus inoculum was added in each well, as well as the negative control (only media), and the 96-well plate was incubated under two different conditions at 24 h at 24˚C and 72 h at 30˚C under aerobic conditions for the biofilm formation. After the establishment of mature biofilms, the TSB with NaCl 1% (w/v) was gently removed from the wells, followed by a gentle wash with PBS (pH 7.4), and biofilm samples were then supplemented with 1x to 32 x MIC values of antibiotics or plant extracts (Eucalyptus and Guava leaves). Plates were again incubated for an additional 24 h at the same previous temperature (24 or 30˚C) under aerobic conditions. Again, in both biofilm assays, a row of TSB with 1% (w/v) NaCl plus pathogen inoculum without any antibiotic or plant extract was used as growth control (or positive control), and a row of wells was filled with medium without inoculum and antibiotic/plant extract was applied to control of sterility (negative control). After finishing the biofilm inhibition and eradication assays, all samples were evaluated for the antimicrobial activity of the antibiotics and plant extracts through the methodologies previously described (biomass quantification and CFU counting assays). Likewise, all the assays were independently performed in triplicate on different days.

## 2.8 Statistical analysis

**2.8.1 Response surface methodology: Experimental design of biofilm formation.** A multilevel multifactorial model was chosen due to the difference in the number of levels between factors to be analyzed. The present model was composed of a 2 x 2 x 2 x 3 factorial design without midpoints. In the experiments, the levels of the factors for the formation of biomass and viability varied, these were carried out according to the multifactorial model. Therefore, only six tests were necessary to determine the influence of the factors on the formation of biomass and viability. In the multifactorial model, there were their levels to be studied (2 *Vibrio* species, 2 temperatures, 2 initial inoculums, and 3 times). The obtained data allow us to evaluate a second-order response surface with an adjusted model illustrating the curvature of

the effects of the factors and their interactions (Eq 1).

$$y = \beta_0 + \sum_{i=1}^{k} \beta_i x_i + \sum_{i=1}^{k} \sum_{j=1}^{k} \beta_{ij} x_i x_j + \sum_{i=1}^{k} \beta_{ij} x_i^2 + \in \qquad (Eq1)$$

**2.8.2 Response surface methodology: Experimental design of biofilm treatment with antibiotics and plant extract.** For the experimental design of the treatments against biofilms, the two best conditions for biofilm formation were selected. Subsequently, the analysis of composite desirability was carried out on the experimental design of the formation of biofilms. A multilevel multifactor model was also performed. This model was composed of a 2 x 3 x 4 factorial design without midpoints. In the experiments, the levels of the factors for the formation of biomass and viability varied and so these data were carried out according to the multifactorial model. In the multifactorial model, 2 optimal training conditions, 3 *Vibrio* species, and 4 treatments were evaluated. The obtained data allow us again to analyze a second-order response surface with an adjusted model evidencing the curvature of the effects of the factors and their interactions (Eq 1).

**2.8.3 ANOVA Multifactorial statistical analysis.** All data in the present study was obtained from at least triplicate assays done on different days. In the case of biomass growth, CFU counting, and total cell or live/dead cell counting, each assay was performed using five samples per assay. All data results were carried out to obtain the mean, minimum, maximum, and standard deviation (SD). The normality data set was analyzed using the Anderson-Darling, Ryan-Joiner, and Kolmogorov-Smirnov normality tests. If the data had a non-normal distribution, nonparametric tests were used. The Johnson data transformation method determines the best value for lambda ($\lambda$), thus obtaining the best type of transformation from non-parametric to parametric data. Once the data were transformed, a multivariate ANOVA analysis was performed and p-values <0.05 were considered significant. To compare the means between the factors of each variable or condition, the Tukey and Fisher tests were performed with an alpha of 0.05 ($\alpha = 0.05$). Likewise, the values obtained between the factors of each variable with $p$ <0.05 are considered significant. Finally, an optimization analysis of the model was carried out based on the composite desirability function to obtain the best combinations of levels and variables evaluating the best growth conditions for the development of biofilms. All data analysis was performed in Minitab version 20 [46].

**2.8.4 Multi-objective optimization and general desirability function.** The desirability model states that for each response ($Y_i$), a desirability function ($d_i(Y_i)$) is necessary to consider whether to maximize, minimize, or achieve an objective [47, 48]. This approach was originally proposed by Harrington (1965) but further improved by Derringer and Suich (1980), where an estimated response is transformed into an unscaled value known as desirability [49]. The data values are assigned between 0 and 1 of these transformed responses $d_i$, which can have many different forms. Regardless of the form, $d_i = 0$ represents a completely undesirable value and $d_i = 1$ symbolizes a completely desirable or ideal response value.

For the desirability transformation, values can be used unilaterally to maximize or minimize $Y_i$ (Eq 2), being the unilateral transformation of maximization used for the desirability analysis; meanwhile, the two-sided transformation is applied to obtain the target value $T_i$ for $Y_i$ where $L_i$ and $U_i$ correspond to the lower and upper limits of the studied response, respectively.

$$d_i = \begin{cases} 0 & \text{if } \hat{Y}_i \leq L_i \\ \left[ \dfrac{\hat{Y}_i - L_i}{T_i - L_i} \right]^s & \text{if } L_i < \hat{Y}_i < T_i \\ 1 & \text{if } \hat{Y}_i \geq T_i = U_i \end{cases} \qquad (Eq2)$$

The exponent *s* corresponds to the weighted factor, which indicates the importance of each response to reach the target value $T_i$. For *s* = 1, the stated desirability function increases linearly up to *Ti*.

Therefore, individual desirability can be matched by employing the observed general desirability function *D(Yi)*. It is defined as the geometric average of the individual desirability functions of each response $d_i$ (*Yi*) (Eq 3), where *n* is the number of responses, instead of directly using the measured (*Yi*) and predicted (*i*) response values.

$$D = (\prod_{i=1}^{n} d_i(Y_i))^{\frac{1}{n}} \tag{Eq3}$$

The optimal solutions are determined by maximizing the observed general desirability D (Yi) through means of the Levenberg-Marquardt or NelderMead algorithm [47], considering the control limits for each variable (S1 Table).

## 3 Results

### 3.1 Identification of chemical compounds in Guava and Eucalyptus leaf extracts

Guava (*Psidium guajava*) and Eucalyptus (*Eucalyptus globulus*) were identified as the plants with the best antibacterial activity in preliminary analyses (data not shown). In addition, Guava (*Psidium guajava*) and Eucalyptus (*Eucalyptus globulus*) showed yield values in the extracts of 19.6% (± 0.6) and 19.7% (± 0.4), respectively. Therefore, these plant extracts were selected, and their chemical composition was analyzed by HPLC-DAD-MS/MS, being the tentative identification shown in Table 1. The ESI was performed in positive ionization mode and several molecules were detected in both extracts. Regarding Guava extract, the identified molecules correspond to the predominant peaks observed in the obtained chromatogram. The first peak at 13.78 was identified as reynoutrin or avicularin (ID1). The secondary rupture of 303 is associated with quercetin and the loss of 132 uma corresponds with a pentoside anglicone. However, it is not possible to differentiate both alternatives with the detected rupture even when the MzCloud provided a higher confidence for reynoutrin (94.4%) than avicularin (88.6%). On the other hand, the 162 loss (ID3) as well as the m/z 303 is consistent with a quercetin glucoside. In the literature, both compounds had been reported in guava leaf extracts [50, 51]. The second compound, puerarin (ID2) is an isoflavone and was identified by MzCloud with 92.4% of similarity. Two anthocyanins were identified, more exactly, cyaniding 3-O-pentoside (ID4), which is characterized by the presence of 287 m/z, and petunidin 3-O-pentoside (ID6), which is identified by the primary product ion at 317 m/z [52]. Finally, for the identification of guavinoside compounds (ID5, ID7, and ID10), we based on previous studies [50, 53]. Moreover, the secondary rupture of 259 m/z in 181 m/z and 105 m/z was identified in MzCloud as 4-benzoylbiphenyl (85.3). The rupture of compounds with ID8 and ID9 was not identified.

Meanwhile, the identified molecules in Eucalyptus leaf extract also correspond to the predominant peaks observed in the chromatogram (Table 1). Its first peak at 14.45 (ID1) was identified as rutin with a similarity in MzCloud of 88.7%. Moreover, the neutral loss of 308 uma is consistent with a loss of glucose and rhamnose as present in rutin. The ID2 and ID9 were identified as luteolin-7-O-betaglucuronide and resinoside by comparing similar ruptures to those presented in Okhlopkova and colleagues' study [54]. Moreover, resinoside A had been previously described in Eucalyptus [55]. The ID4, ID5, and ID6 compounds were identified as quercetin 3-O-glucuronide, 6-O-methylscutellarin, and glycitin by similarity in the MzCloud database with the following scores of 87.9%, 75.8%, and 90.7%, respectively. The ID7 compound was tentatively identified as cypellogin A or B by similar ruptures in the Eucalyptus plant as

**Table 1.** Metabolites identified in Guava (*Psidium guajava* L.) and Eucalyptus (*Eucalyptus globulus* Labill.) aqueous leaf extracts by HPLC-ESI-MS/MS in positive ionization mode.

| ID | RT (min) | m/z [M+H]+ | Fragmentation | Tentative identification |
|---|---|---|---|---|
| colspan | | | Metabolites identified in Guava (*Psidium guajava* L.) extract | |
| 1 | 13.78 | 435 | 303(100)<br>303-> 257(100), 165(80), 285(75), 229(50) | Reynoutrin or Avicularin |
| 2 | 13.94 | 417 | 399(100), 351(30), 381(25), 297(15) | Puerarin |
| 3 | 14.47 | 465 | 303(100) | Quercetin glucoside |
| 4 | 15.31 | 419 | 287(100), 257(10) | Cyanidin 3-O-pentoside |
| 5 | 15.84 | 587 | 285(100), 303(20) | Guavinoside C |
| 6 | 16.16 | 449 | 317(100), 287(50) | Petunidin-3-O-pentoside |
| 7 | 16.35 | 545 | 527(100), 315(40), 273(30), 387(20) | Guavinoside A isomer |
| 8 | 17.0 | 439 | 261(100), 421(10), | NI |
| 9 | 17.3 | 439 | 369(100), 277(15), 259(10) | NI |
| 10 | 17.9 | 573 | 259(100), 315(60), 297(20), 555(15), 181(10)<br>259-> 181(100), 105(15) | Guavinoside B |
| colspan | | | Metabolites identified in Eucalyptus (*Eucalyptus globulus* Labill.) extract | |
| 1 | 14.45 | 611 | 303(100), 465(35), 593(10) | Rutin |
| 2 | 14.50 | 463 | 287(100) | Luteolin-7-O-betaglucuronide |
| 3 | 14.50 | 549 | 429(100), 379(80), 277(50), 361(45), 331(40), 531(35) | NI |
| 4 | 14.55 | 479 | 303(100), 317(25) | Quercetin 3-O-glucuronide |
| 5 | 16.42 | 477 | 301(100), 315(15), 459(10) | 6-O-methylscutellarin |
| 6 | 16.43 | 447 | 285(100) | Glycitin |
| 7 | 18.11 | 631 | 303(100), 311(75), 613(55), 329(35), 595(20) | Cypellogin A or B |
| 8 | 18.40 | 487 | 469(100), 311(60), 451(35)<br>311-> 149(100), 293(75), 167(15) | Like cypellocarpin C (see text) |
| 9 | 19.31 | 615 | 287(100), 311(60), 597(35), 329(30) | Resinoside–A |
| 10 | 19.90 | 521 | 503(100), 485(20)<br>503-> 193(100), 311(40), 293(20),<br>167(15), 485(10) | Cypellocarpin C |
| 11 | 20.44 | 563 | 545(100)<br>545-> 425(100), 259(95), 301(65), 379(60), 527(55), 361(50), 191(45) | NI |

NI: not identified.

reported by Boulekbache-Makhlouf and colleagues [56]. The ID8 compound is similar to cypellocarpin C (ID10), where the neutral loss of water (18 uma) produces the product ion at 469 m/z with a further loss of 158 uma probably associated with a consecutive loss of water (leading to 451 m/z), and 140 uma producing the ion at 311 m/z. This rupture could suggest a ring with 3-hydroxyl substitution like in the gallic acid [57]. All these compounds are common in Eucalyptus extracts by previous studies [58, 59], as well as cypellocarpin C (ID10) was identified in the MzCloud database with a score of 73.7%. It is important to mention that the reason for the "low" similarity could be due to the consequence of the second rupture. Therefore, the MzCloud peaks were mainly identified through the second rupture derived from the first one. Finally, the chemical compounds labeled ID11 and ID3 were not identified based on their rupture patterns.

### 3.2 Quantification of *V. parahaemolyticus* and *V. cholerae* biofilms and their normality assessment

The ability of *V. parahaemolyticus* strain VP 87 and *V. cholerae* strain VC 112 to develop a biofilm was determined by comparing biomass, cell viability, and total cell and live/dead cell

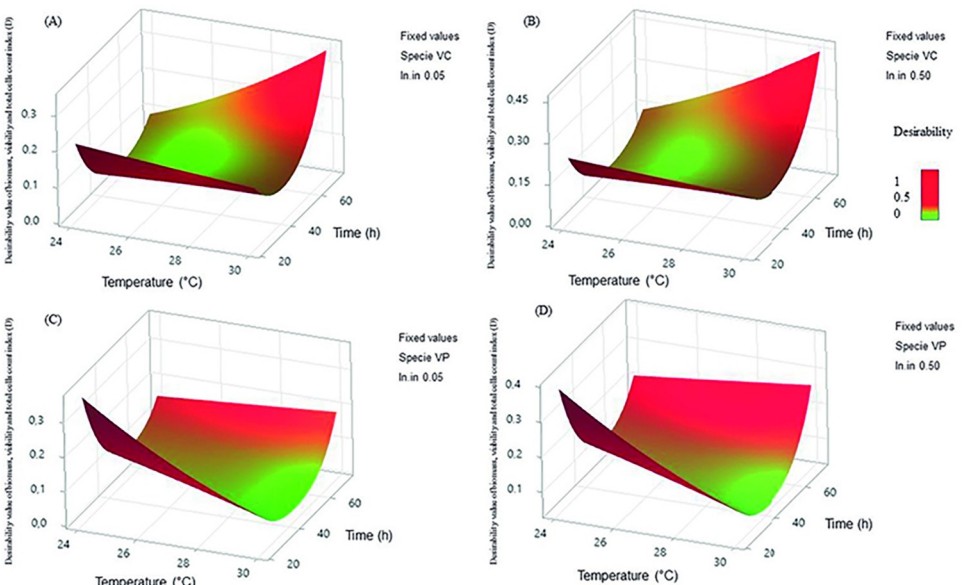

**Fig 1. The response surfaces and estimated contours of the desirability value of *Vibrio* spp. biofilm formation under different growth conditions to determine MIC.** Response surface for the interaction between the desirability of biofilm formation, based on the 3D plot of all combinations of four variables keeping two fixed variables (*Vibrio* species and initial inoculum) and two continuous variables (temperature and time) for evaluation. (a) The desirability of *Vibrio cholerae* biofilm formation with an initial inoculum of 0.05 McFarland versus temperature (24°C and 30°C) and time (24, 48, and 72 h). (b) The desirability of *Vibrio cholerae* biofilm formation with an initial inoculum of 0.50 McFarland versus temperature (24°C and 30°C) and time (24, 48 and 72 h). (c) The desirability of *Vibrio parahaemolyticus* biofilm formation with an initial inoculum of 0.05 McFarland versus temperature (24°C and 30°C) and time (24, 48 and 72 h). (d) The desirability of *Vibrio parahaemolyticus* biofilm formation with an initial inoculum of 0.50 McFarland versus temperature (24°C and 30°C) and time (24, 48, and 72 h).

counting assays (Fig 1 and S2 and S3 Tables). An assessment of normality was also applied to the obtained data, using the Anderson-Darling, Ryan-Joiner, and Kolmogorov-Smirnov normality test and the Johnson data transformation method. Subsequently, a reduction of the variable interaction model was performed, eliminating interactions that showed a $p > 0.05$ (S4 Table). Finally, the same statistical analysis was repeated in the new data set.

As shown in Fig 1 and S3 Table, *V. parahaemolyticus* biofilms showed a higher ability to produce biomass and culturable sessile cells (CFU counting) during the first 24 h when compared to *V. cholerae* biofilms ($p < 0.001$). However, at 72 h, *V. cholerae* biofilms had a higher ability to produce biomass and culturable sessile cells. Additionally, *V. cholerae* biofilms showed a greater ability to produce biomass and culturable sessile cells at 30°C when compared to 24°C ($p < 0.001$). Normality tests showed normal distribution after the reduction of the model in biomass and viability tests, but a non-normal distribution in the total cell count test, so a data transformation by Johnson's method was performed. Consequently, a parametric statistical analysis was selected for future evaluation.

### 3.3 Comparison between growth conditions in *Vibrio*-associated biofilms

Biomass and viable cells within the biofilm were quantified demonstrating statistical differences between species at different temperatures, times, and initial inoculums (see Fig 2 and S4 Table). Multivariate ANOVA analysis showed a significant effect of the variables and the interactions with a $p < 0.05$, to determine the biomass growth and biofilm viability in both *Vibrio* species (see S4 Table). Both *Vibrio* biofilms had a significant increase in biomass and cell

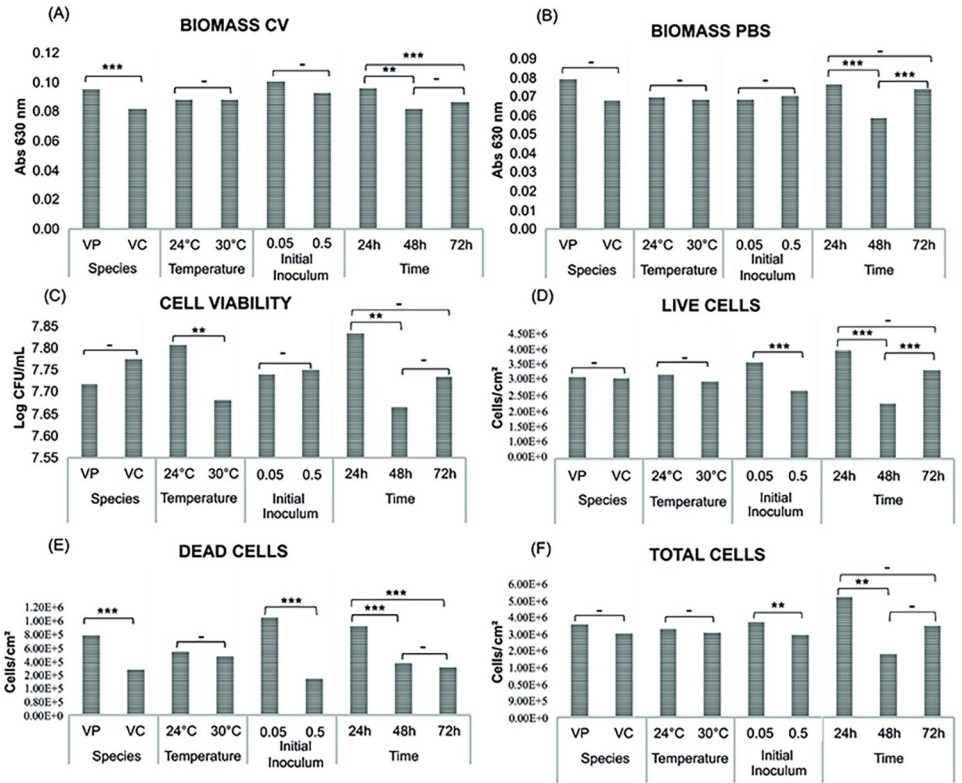

**Fig 2. Statistical analysis of the growth conditions involved in the biofilm formation of *V. parahaemolyticus* and *V. cholerae*.** The variables determined for biofilm formation of *V. parahaemolyticus* (VP) and *V. cholerae* (VC) were temperature (24 and 30°C), initial inoculums (0.05 and 0.5 McFarland), and time (24, 48, and 72h). The biomass of the biofilm was analyzed by two methodologies measuring the optical density by absorbance at 630 nm with crystal violet and PBS suspension, while cell viability analysis was evaluated by colony-forming counting assays, and biofilm quantification was evaluated by total cell and live/dead cells assays. These results were analyzed using the Tukey and Fisher tests methodology for growth conditions comparison, where the bars show the mean of the variables for each result, in order to determine whether or not there is a significant difference between the variables (growth conditions). (a) Biofilm biomass by crystal violet (CV) staining method, (b) Biofilm biomass by phosphate-buffered saline (PBS) suspension method, (c) Biofilm viability by colony-forming (CFU) counting assays, (d) Biofilm quantification by live cell count with Live/Dead staining, (e) Biofilm quantification by dead cell count with Live/Dead staining, and (f) Biofilm quantification by total cell count with Live/Dead staining. * $p < 0.05$; ** $p < 0.01$; *** $p < 0.001$; **** $p < 0.0001$, —non-significant.

viability at 24 h of growth when compared to 48 h with a $p < 0.001$ in both species by Tukey and Fisher tests, where the biomass and cell viability of *Vibrio*-related biofilms significantly decreased by 43% and 0.5 log (see Fig 2), respectively. Interestingly, *Vibrio*-associated biofilms showed a significant increase in biomass and cell viability at 72 h (see S4 Table). However, biomass and cell viability had a higher growth of *V. parahaemolyticus* biofilm at 24°C, while *V. cholerae* biofilm showed a higher growth at 30°C, as previously observed through desirability analysis. It is worth mentioning that the initial inoculum was not significant for cell viability ($p = 0.45$), but it was significant for biomass growth ($p < 0.001$) with a slight augmentation of almost 10% at 1E+8 CFU (0.5 MacFarland) in both *Vibrio* species biofilms, therefore it was selected for further biofilm inhibition and eradication assays.

The statistical analysis of biofilm growth by variables comparison method with Tukey and Fisher tests showed a significant difference between *V. parahaemolyticus* and *V. cholerae* biofilms in the biomass assays with crystal violet staining ($p < 0.001$; S5 Table), but the biomass assays with PBS suspension and cell viability assay did not show any significant differences

($p$ = 0.20–0.26), so biomass assays with PBS suspension were chosen for further biofilm inhibition and eradication assays. Likewise, in both *Vibrio* species, Tukey and Fisher tests confirmed statistical differences in all biofilm samples during time in the biomass growth and cell viability assays, in particular between 24 and 48 h ($p < 0.001$). Contrasting with the previous multivariate ANOVA analysis, Tukey and Fisher tests did not show any significant difference for the initial inoculum variation in the biomass growth and cell viability assays for both species ($p$ = 0.10–0.86). Although the temperature in the biomass assays did not evidence significant differences ($p$ = 0.51–0.99), Tukey and Fisher tests demonstrated a significant difference for the cell viability assays ($p$ = 0.02). It is interesting to observe that biomass growth trends demonstrating their highest biomass growth and cell viability in the first 24 h at 24°C (Fig 2), where *V. cholerae* reached close to the same level of biomass at 72 h and 30°C (S5 Table) when compared to the conditions of 24 h at 24°C, which contrasted with multivariate ANOVA and desirability analyses.

### 3.4 Total cell and live/dead cells count in *V. parahaemolyticus* and *V. cholerae* biofilms

Next, we analyzed the number of total cells and live/dead cells within biofilms among *Vibrio* species under different growth conditions. As shown in S5 Table, the fluorescence microscopy (FM) analysis showed a significant effect between samples according to temperature, time, and initial inoculum in both *Vibrio* species through the evaluation of total count, live and dead cells within biofilm samples. When comparing species, statistical differences were observed between *Vibrio* species. More exactly, *V. parahaemolyticus* showed a higher number of dead cells within the biofilm (9.09E+5 dead cells/cm$^2$) and approximately 21% of the total cells, while *V. cholerae* only showed 13% of dead cells (5.05E+5 dead cells/cm$^2$) within the biofilm ($p < 0.001$). Although no statistical differences were found between *Vibrio* species in the total count and live cells, both *Vibrio* biofilms evidenced significant differences in their intrinsic growth between 24 and 48 h ($p < 0.001$), as well as 48 and 72 h ($p < 0.001$). Furthermore, concerning temperature variation between 24 and 30°C, both *Vibrio*-related biofilm species did not show statistical differences. Interestingly, in FM analysis, the initial inoculum of both *Vibrio* species for biofilm formation showed significant differences for total cell and live/dead cells, where the initial inoculum of 1E+8 CFU (0.5 MacFarland) demonstrated a higher number of live cells and consequently, a higher number of dead cells for the initial inoculum of 1E+7 CFU (0.05 MacFarland) was observed ($p < 0.05$). Finally, it was also observed that *V. parahaemolyticus* and *V. cholerae* demonstrated a higher number of total cells and live cells within the biofilms at 24 h, 24°C, and 1E+8 CFU (0.5 McFarland), as shown in Fig 3.

Although a thoughtful optimization of the FM analysis was performed during the present study, the pictures of live/dead cells within the biofilms of both species did not show the best clarity. However, the merged images showed a greater clarity of the biofilms' development, and the macros Biofilms Viability checker software provided a reliable assessment of the data set. Visual differences in the percentage of dead and alive cells were detected between *Vibrio* species during time, temperature, and initial inoculum. These differences are also validated by multivariate ANOVA analysis and the optimized growth conditions were provided by desirability analysis (Fig 1) through the weighted geometric mean of the individual desirability of the results by PBS at 630 nm, CFU counting and total cell count responses (see S4 Table). Minitab evaluation determines the optimal settings for the variables by maximizing the composite desirability, showing a homogeneous distribution of dead cells within the biofilms of both *Vibrio* species (see merged images in S2 Fig).

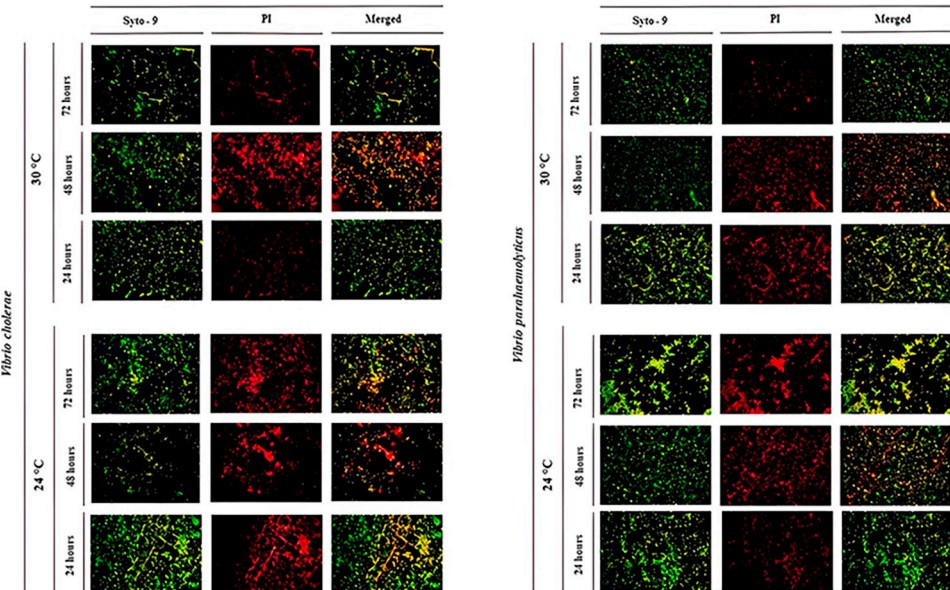

**Fig 3. Illustration of fluorescence microscopy analysis in the biofilm formation of *V. parahaemolyticus* and *V. cholerae* at different growth conditions.** Biofilms of *V. parahaemolyticus* and *V. cholerae* at 24 and 30˚C during time (24, 48, and 72h) by fluorescence microscopy using Live/Dead staining assays. The illustrated biofilm samples with an initial inoculum of 1E+8 CFU (0.5 McFarland) were used to compare the total number of live and dead cells. An Olympus BX50 microscope with 100X magnification was used (scale bar of 5 μm), images were obtained with AmScope software, and images were merged with Fiji-ImageJ software.

## 3.5 Comparison of the four methodologies to assess biofilm development

After the statistical analysis of the data set, we compared the data accuracy of these different and classical methodologies for *Vibrio*-related biofilm characterization. Therefore, multivariate linear regression ANOVA was applied using CFU counting assays as a reference due to its well-known reliability, and R-squared values were observed as a goodness-of-fit measure for biofilm analysis. As expected, fluorescence microscopy analysis revealed the lowest R-squared values for both *Vibrio*-related biofilms ($R^2$ = 0.22–0.45), followed by biomass assays with CV staining ($R^2$ = 0.84–0.85), and finally biomass assays with PBS suspension ($R^2$ = 0.86–0.92). It is important to mention again that all assays were analyzed at least six assays with five biofilm samples, under the same conditions, to prevent data variability, bias, and manual errors by the researcher during biofilm analysis.

## 3.6 Antibacterial planktonic activity of Eucalyptus and Guava leaf extracts

As shown in Fig 4, the inhibition of the planktonic growth of *Vibrio parahaemolyticus* (VP87 and tetracycline-resistant VP275) and *V. cholerae* strains (VC112) by Eucalyptus and Guava leaf extracts was studied using broth microdilution method during 24 h of incubation at 37˚C and compared to tetracycline and ceftriaxone antibiotics. These assays allow us to determine the lowest inhibitory concentration of the antimicrobial agent (minimal inhibitory concentration, MIC) and to further evaluate other extract concentrations gradually inhibiting bacterial growth through bacterial turbidity. Visible inhibition growth of all *Vibrio* isolates was definitively observed at 1600 μg/mL of Eucalyptus extract and 200 μg/mL of Guava extract, being these concentrations considered as MIC values for further biofilm inhibition and eradication assays. To better analyze the variables influencing the planktonic growth of *Vibrio* isolates,

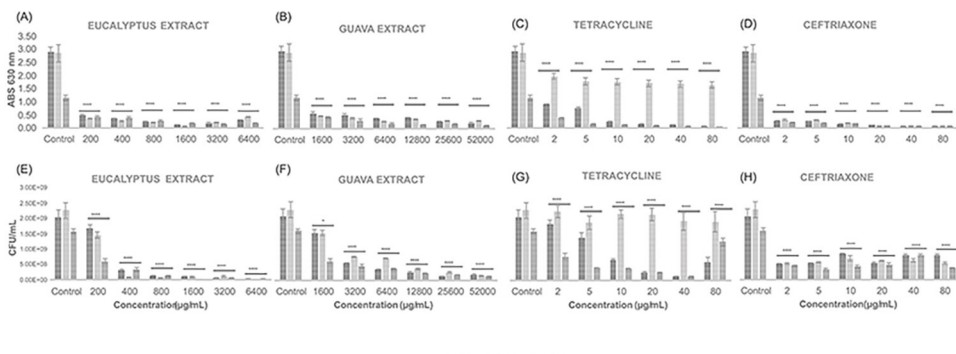

**Fig 4.** Inhibition of growth of *V. parahaemolyticus* and *V. cholerae* strains (VP87, tetracycline-resistant VP275, and VC112) by Eucalyptus (*a*) and Guava (*b*) extracts as well as tetracycline (*c*) and ceftriaxone (*d*) antibiotics at six given concentrations using micro-broth dilution assay in TSB supplemented with NaCl 1% after 24 h of incubation at 37˚C. Bacterial turbidity was measured at OD = 630 nm and MIC values were determined for each plant extract and antibiotic against these planktonic *Vibrio* strains. The illustrated statistical analysis between biomass and treatment concentrations, for each Vibrio species. Using the Tukey and Fisher variable comparison methods to create confidence intervals for all pairwise differences in the means of the variable levels, where the bars show the average of the variables for each outcome, in order to determine whether or not there is a significant difference between variables (Control against treatment concentrations), more specifically * $p < 0.05$; ** $p < 0.01$; *** $p < 0.001$; **** $p < 0.0001$.

statistical comparison tests were conducted using the Tukey and Fisher methods between the positive control (medium plus bacterial inoculum) and the different treatment concentrations.

Eucalyptus leaf extract evidenced statistically significant inhibition at a concentration of 50 μg/mL with an average inhibition of 35% against all *Vibrio* isolates ($p < 0.0001$) and reaching its maximum inhibition at 200 μg/mL with an average inhibition of 75% ($p < 0.0001$). Meanwhile, guava leaf extract started exhibiting a more significant inhibition at a concentration of 400 μg/mL with an average inhibition of 45% against all *Vibrio* isolates ($p < 0.0001$) demonstrating its maximum inhibition at 1600 μg/mL with an average inhibition of 70% ($p < 0.0001$). On the other hand, antibiotic assays exhibited greater inhibition in the growth of planktonic *Vibrio* isolates as expected due to the purified active compounds, and both antibiotics confirmed their MIC values at 2 μg/mL with an average inhibition of 85% against VP87 and VC112 isolates ($p < 0.0001$). However, as previously reported, VP275 isolate exhibited resistance to all tetracycline concentrations but showed a MIC value of 2 μg/mL by ceftriaxone. These results allowed us to further evaluate the antimicrobial activity through biofilm inhibition and eradication assays with these plant extracts and compared with these standard antibiotics using 1x MIC values as a starting point of evaluation.

### 3.7 Biofilm inhibition and eradication assays

Both biofilm inhibition and eradication assays were performed by the two sets of conditions found for the *V. parahaemolyticus* (VP87 and VP275) and *V. cholerae* (VC112) following the desirability analysis for optimal biofilm growth conditions (Fig 1), more exactly, 24 h at 24˚C and 72 h at 30˚C, respectively, before the antibiofilm evaluation by antibiotic and plant extracts. According to the data accuracy analysis of methodologies for *Vibrio*-related biofilm characterization, CFU counting assays and biomass assays with PBS suspension were selected. As shown by Fig 5, both antibiotics and plant extracts were able to inhibit biofilm biomass development starting by 1x MIC values during 24 h at 24˚C, excepting tetracycline against *V. parahaemolyticus* VP275 due to its intrinsic antimicrobial resistance but still showing statistical inhibition when compared to control ($p < 0.0001$). Meanwhile, Guava extract showed its higher biomass inhibition (87%) at the concentration of 12800 μg/mL (8x MIC value) and 90%

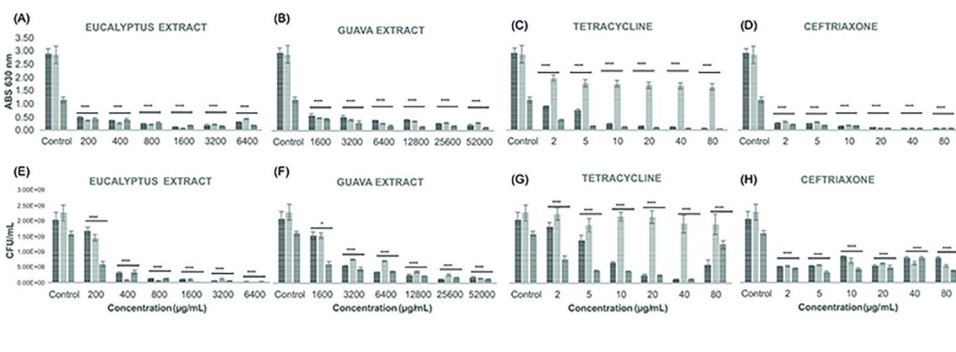

**Fig 5.** Illustration of the biomass evaluation at 630 nm and colony-forming units (CFU) counting in the biofilm inhibition assays of *V. parahaemolyticus* and *V. cholerae* strains (VP87, tetracycline-resistant VP275, and VC112) by Eucalyptus (*a* and *e*) and Guava (*b* and *f*) extracts as well as tetracycline (*c* and *g*) and ceftriaxone (*d* and *h*) antibiotics at six given concentrations in TSB supplemented with NaCl 1% during 24 h of incubation at 24°C. Bacterial turbidity was measured at OD = 630 nm and MIC values were determined for each plant extract and antibiotic against these planktonic Vibrio strains. The illustrated statistical analysis between biomass and treatment concentrations, for each Vibrio species. Using the Tukey and Fisher variable comparison methods to create confidence intervals for all pairwise differences in the means of the variable levels, where the bars show the average of the variables for each outcome, in order to determine whether or not there is a significant difference between variables (Control against treatment concentrations), more specifically * $p < 0.05$; ** $p < 0.01$; *** $p < 0.001$; **** $p < 0.0001$.

of inhibition in viable cells at the concentration of 25600 μg/mL (16x MIC value) against all *Vibrio* sp. biofilms ($p < 0.0001$; see S6 Table). However, differences were observed in the CFU counting between antibiotics and plant extracts, where both extract plants demonstrated a higher reduction of viable cells when compared with both antibiotics at 8x, 16x, and 32x MIC values (Eucalyptus extract: 1600, 3200, and 6400 μg/mL; while Guava extract: 12800, 25600, and 52000 μg/mL). Concerning the biofilm growth conditions of 72 h at 30°C, different results were obtained as shown in Fig 6, where the inhibition of biofilm biomass formation was less notorious by Guava leaf extract and tetracycline although still maintaining statistical differences when compared to control ($p < 0.0001$). When looking at the biomass formation, ceftriaxone apparently evidenced a higher *Vibrio*-related biofilm inhibition when compared to the Eucalyptus extract at all MIC values. However, CFU counting assays demonstrated that, despite a higher biomass presence within the biofilm samples, Eucalyptus extract notoriously reduced the total number of viable cells within *Vibrio*-biofilm samples from 2x until 32x MIC values (400–6400 μg/mL) when compared to the same MIC values of ceftriaxone (5–80 μg/mL), which was not able to reduce the number of viable cells after 1x MIC value (2 μg/mL). Moreover, Eucalyptus extract demonstrated similar results with no significant differences in terms of biomass and CFU reduction in biofilm inhibition assays at both sets of growth conditions (24 h at 24°C and 72 h at 30°C), showing an average inhibition of approximately 80% at the concentration of 400 μg/mL for all *Vibrio* isolates ($p < 0.0001$; see S7 Table).

A further antibiofilm evaluation was studied through eradication biofilm assays and remarkably the same trend of results was observed, i.e., significant eradication values were obtained against all *Vibrio* species or strains starting by 1x MIC values at both sets of growth conditions (24 h at 24°C and 72 h at 30°C) when compared to control ($p < 0.0001$), as shown in Figs 7 and 8. However, as expected, biofilm eradication values were substantially lower by all antibiotics and plant extracts when compared to biofilm inhibition assays. Once again, differences were detected in the CFU counting between antibiotics and plant extracts, where both extract plants demonstrated higher reduction of viable cells when compared with both antibiotics at 8x, 16x, and 32x MIC values at both sets of growth conditions except for *V. cholerae*

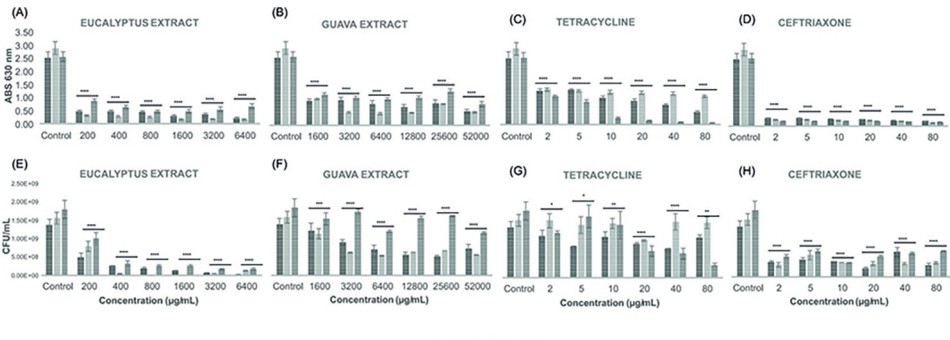

**Fig 6.** Illustration of the biomass evaluation at 630 nm and colony-forming units (CFU) counting in the biofilm inhibition assays of *V. parahaemolyticus* and *V. cholerae* strains (VP87, tetracycline-resistant VP275, and VC112) by Eucalyptus (*a* and *e*) and Guava (*b* and *f*) extracts as well as tetracycline (*c* and *g*) and ceftriaxone (*d* and *h*) antibiotics at six given concentrations in TSB supplemented with NaCl 1% during 72 h of incubation at 30˚C. The illustrated statistical analysis between biomass and treatment concentrations, for each *Vibrio* species. Using the Tukey and Fisher variable comparison methods to create confidence intervals for all pairwise differences in the means of the variable levels, where the bars show the average of the variables for each outcome, in order to determine whether or not there is a significant difference between variables (Control against treatment concentrations), more specifically * $p < 0.05$; ** $p < 0.01$; *** $p < 0.001$; **** $p < 0.0001$.

(VC112) biofilms of 72 h at 30˚C (Fig 8), which demonstrated resistance against Guava extract in both biomass and cell viability evaluation thus confirming its optimal growth conditions previously assessed. Nonetheless, Guava extract demonstrated excellent eradication values in biofilms of 24 h at 24˚C reducing 60% of biomass and 90% of viable cells at the concentration of 25600 μg/mL for all *Vibrio* isolates ($p < 0.0001$; see S7 Table), while 72 h and 30˚C exhibit its greater effectiveness with an average inhibition of 72% in biomass and viable cells at the concentration of 51200 μg/mL against *Vibrio* isolates ($p < 0.0001$; see S7 Table). Meanwhile, Eucalyptus extract at 800 μg/mL reduced approximately 70% of biomass and 90% of viable cells for all *Vibrio* isolates ($p < 0.0001$). It is important to mention that no significant difference was observed in the eradication ability of Eucalyptus extract at both sets of growth

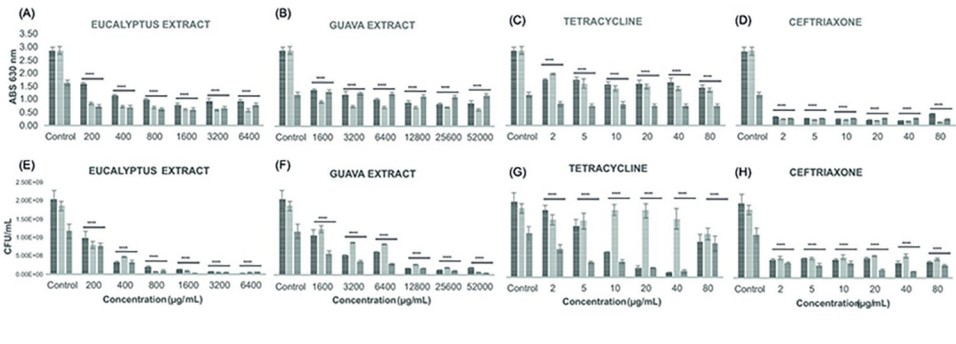

**Fig 7.** Illustration of the biomass evaluation at 630 nm and colony-forming units (CFU) counting in the biofilm eradication assays of *V. parahaemolyticus* and *V. cholerae* strains (VP87, tetracycline-resistant VP275, and VC112) by Eucalyptus (*a* and *e*) and Guava (*b* and *f*) extracts as well as tetracycline (*c* and *g*) and ceftriaxone (*d* and *h*) antibiotics at six given concentrations in TSB supplemented with NaCl 1% after 24 h of incubation at 24˚C. The illustrated statistical analysis between biomass and treatment concentrations, for each *Vibrio* species. Using the Tukey and Fisher variable comparison methods to create confidence intervals for all pairwise differences in the means of the variable levels, where the bars show the average of the variables for each outcome, in order to determine whether or not there is a significant difference between variables (Control against treatment concentrations), more specifically * $p < 0.05$; ** $p < 0.01$; *** $p < 0.001$; **** $p < 0.0001$.

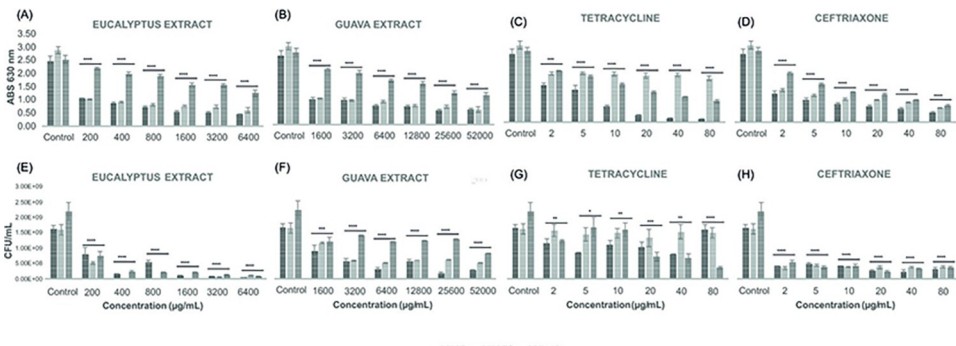

**Fig 8.** Illustration of the biomass evaluation at 630 nm and colony-forming units (CFU) counting in the biofilm eradication assays of *V. parahaemolyticus* and *V. cholerae* strains (VP87, tetracycline-resistant VP275, and VC112) by Eucalyptus (*a* and *e*) and Guava (*b* and *f*) extracts as well as tetracycline (*c* and *g*) and ceftriaxone (*d* and *h*) antibiotics at six given concentrations in TSB supplemented with NaCl 1% after 72 h of incubation at 30°C. The illustrated statistical analysis between biomass and treatment concentrations, for each *Vibrio* species. Using the Tukey and Fisher variable comparison methods to create confidence intervals for all pairwise differences in the means of the variable levels, where the bars show the average of the variables for each outcome, in order to determine whether or not there is a significant difference between variables (Control against treatment concentrations), more specifically * $p < 0.05$; ** $p < 0.01$; *** $p < 0.001$; **** $p < 0.0001$.

conditions apart from the biomass eradication of *V. cholerae* (VC112) biofilms of 72 h at 30°C (Fig 8). Therefore, the overall results of the present study demonstrated the higher effectiveness of Eucalyptus extract to inhibit and eradicate *Vibrio*-related biofilms independently of time and temperature conditions.

## 4 Discussion

The *Vibrio* genus possesses several pathogens that directly affect the aquaculture industry, according to the Food and Agriculture Organization (FAO), such as the shrimp industry [2]. Nowadays, the shrimp industry is experiencing the highest growth rate with production increasing by approximately 9.4% annually [1]. However, chemical products have been used for decades to treat *Vibrio*-associated infections, leading to an increase in antimicrobial resistance [60]. These *Vibrio*-associated infections result in economic losses of up to 3 billion US dollars in global production, affecting all countries that produce crustaceans [23, 61]. In Ecuador, the shrimp industry represents 19% of its exportation market and 5 billion US dollars. According to the Global Outlook for Aquaculture Leadership (GOAL) in 2019, Ecuador is the leading shrimp producer and exporter in the world [3]. As a result, shrimp production losses have a significant impact on the Ecuadorian economy. Therefore, efforts have been made to find an eco-friendly solution using plant extracts to counteract the problems of resistance and environmental impact associated with antibiotic use, which is prohibited in Ecuador and other countries, or other chemical compounds. In addition, *V. parahaemolyticus* and *V. cholerae* are commonly found in shrimps at retail, farms, and estuaries until nowadays constituting an eminent and serious Public Health risk to the Ecuadorian population and an alarming economic threat to the shrimp industry, as evidenced by Clydent S.A. and other authors [25, 26].

In the present study, we first characterized the main variables (temperature, time, and initial inoculum) influencing the development of biofilms by *V. parahaemolyticus* and *V. cholerae*. The initial inoculum was not significant for cell viability, but it was significant for biomass growth with a slight augmentation at 1E+8 CFU in both *Vibrio* species biofilms, as expected from studies with different biofilm-forming microorganisms [31, 62, 63]. However, our results evidenced that optimized *V. parahaemolyticus* biofilms were obtained at 24°C

during 24 h while optimized *V. cholerae* biofilms were observed at 30°C during 72h. However, it is important to mention that, during the realization of biofilm assays, *V. parahaemolyticus* evidenced thicker biofilms during the first 24 h but *V. cholerae* biofilms showed more resilience to the washing steps during time (72 h) which could influence the results when comparing the four methodologies to assess biofilm development. Due to the observed biofilm thickness in both *Vibrio* species, the fluorescence microscopy analysis through LIVE/DEAD assays could be affected by the dye infiltration (SYTO-9 and PI) within the samples resulting in a lower R-squared value ($R^2$ = 0.22–0.45) for biofilm analysis. Although the LIVE/DEAD staining assay was previously optimized (data not shown), the present methodology requires to be combined with other methodologies, as previously observed [64, 65]. Also, this initial characterization agreed with previous studies [17, 19], confirming that certain growth variables could influence the *Vibrio* species' dominance in the aquatic environment and *Vibrio*-associated biofilm infections, as well as different genetic controls of biofilm formation by *V. parahaemolyticus* and *V. cholerae*. In the present work, it was possible to observe the differences in biofilm regulation between *V. cholerae* and *V. parahaemolyticus* involving distinctive transcriptional *vps* and *cps* genes through time and temperature [13, 19], respectively. Furthermore, several authors already reported the great diversity of *Vibrio* species in seafood, including shrimp, but also in different geographical areas [66–68]. In fact, a systematic review and meta-analysis demonstrated that *V. parahaemolyticus* could only be isolated around 47.5% of the 5811 seafood samples analyzed [14], evidencing the heterogeneity and prevalence of *Vibrio* species among seafood. Therefore, the present work also showed the importance of expanding surveillance on different *Vibrio* species among seafood types and industrial aquaculture areas to identify critical control points (such as temperature and time) for the seafood production system and avoid public health impact.

Several studies have been conducted with alternative treatments to control *Vibrio*-associated infections using probiotic and natural-derived products. In Thailand, Yatip and colleagues applied extract from the fermented soybean product Natto to inhibit *Vibrio* biofilm formation [28], while Karnjana and colleagues evaluated the ethanolic extract of the red seaweed *G. fisheri* suggesting its future application as an alternative treatment for biofilm-associated *Vibrio* infections in aquacultures [29]. While Kamble and colleagues assessed the antimicrobial activity of sulphated galactans of *G. fisheri* against *V. parahaemolyticus* and *V. harveyi* [30]. All these studies analyzed abundant and native natural products of Thailand's resources for their application in the shrimp industry. Likewise, Ecuador evidenced abundant and natural products with well-known antimicrobial activity such as Eucalyptus (*Eucalyptus globulus*) and Guava (*Psidium guajava*) [60, 69–72], which evaluated the antimicrobial activity of Eucalyptus and Guayaba extracts against several *Vibrio* species (including *V. parahaemolyticus*). However, to the authors' best acknowledge, none of the previous studies characterized biofilm formation of *V. parahaemolyticus* and *V. cholerae* under different growth variables nor evaluated cell viability after biomass assays. As demonstrated in this study, the cell viability reduction was significantly higher when compared to biomass assays and antibiotic activity, thus illustrating the novelty of the present study. Therefore, the planktonic antibacterial activity of Eucalyptus and Guava extracts was evaluated against *V. parahaemolyticus* and *V. cholerae* strains and compared to antibiotic treatments. Our results revealed that Eucalyptus extract exhibited a significant planktonic growth inhibition on these *Vibrio* species at concentrations ranging from (50 to 200 μg/mL), being similar values to those reported by Sivaraj and colleagues [73] for *V. parahaemolyticus* (31.25 to 125 μg/mL), as well as Nwabor and colleagues [74] for *Listeria monocytogenes* (64 to 128 μg/mL). On the other hand, the Guava extract displayed a planktonic growth inhibition in the range of 400 to 1600 μg/mL in this study, which are lower values than the values reported by Rahim et al. [75] and Hoque et al. [71], more

exactly concentrations between 1250 and 3000 μg/mL for *V. cholerae* and *V. parahaemolyticus.* In comparison, tetracycline and ceftriaxone exhibited higher inhibition at 2 μg/mL, as purified compounds, in sensitive strains (VP87 and VC112) but were unable to inhibit resistant *Vibrio* strains such as tetracycline-resistant VP275.

Further evaluation was performed through biofilm inhibition and eradication assays using biomass and CFU counting methodologies. To the authors' best knowledge, the present study is the first study to report the biofilm inhibition and eradication assays against *Vibrio* species with both methodologies, allowing thus to characterize biofilm reduction and cell viability within biofilms. Given the absence of literature about cell viability assays, our results were compared with those studies evaluating biomass reduction. However, it is important to note that plant extracts revealed equal (Guava extract) or greater (Eucalyptus extract) biofilm inhibition and eradication values by CFU counting evaluation when compared to tetracycline and ceftriaxone, evidencing greater antibiofilm activities than the initial biomass evaluation and also suggesting synergy between multiple compounds in both plant extracts. Regarding biomass evaluation, Eucalyptus extract achieved a 90% of biofilm inhibition at 400 μg/mL (2x MIC value). Although no studies were found regarding biofilm inhibition and eradication in *Vibrio* species, several studies reported biofilm inhibition levels up to 95.9% with Eucalyptus extract ranging from 250 to 393 μg/mL against other bacterial species such as *L. monocytogenes*, *Escherichia coli*, *Staphylococcus aureus*, and *Streptococcus pyogenes* [74, 76, 77]. Concerning biofilm eradication assays, Eucalyptus extract at 800 μg/mL (4x MIC value) was able to eradicate more than 70% of all *Vibrio* biofilms in agreement with Tsukatani and colleagues where Eucalyptus extract eradicated 99% of *Porphyromonas gingivalis* JCM12257, *S. aureus* NBRC13276, and *Streptococcus mutans* NBRC13955 biofilms showing minimum biofilm eradication concentration (MBEC) values of 49.1 μg/mL for *P. gingivalis* and 393 μg/mL for *S. aureus* and *S. mutans* [78]. However, in the same study, Eucalyptus extract at superior concentrations of 1572 μg/mL was only able to eradicate 21.0% of *E. coli* NBRC15034 biofilms [78]. Meanwhile, in the present study, the Guava extract showed a higher biomass inhibition at concentrations of 12800 μg/mL (8x MIC value), evidencing an average of 80% biomass inhibition among *Vibrio* isolates except for VC112 at 30°C. Further evaluation demonstrated the Guava extract's ability at 25600 μg/mL (16x MIC value) to eradicate an average of 70% of the biomass in all *Vibrio* sp. excepting once again VC112 at 30°C. In terms of eradication, other studies only evaluated Guava extract against different pathogenic species, such as *S. aureus* [79, 80]. These studies reported eradication levels of up to 44% and 59% in *S. aureus* biofilms when applying Guava extracts from 440 to 870 μg/mL and 1100 to 2100 μg/mL [79, 80], respectively. These lower eradication values when compared to the present study could be due to the lower applied concentrations of Guava extract as well as the nature of *S. aureus* biofilms virulence mechanisms. Despite the use of different measurement methods, our MIC values agree with the values reported by Sivaraj et al. and Nwabor et al. showing MIC values for Eucalyptus extract of 31.25 to 125 μg/mL and 64 to 128 μg/mL [73, 74], respectively. Likewise, Rahim et al. and Hoque et al. also reported similar MIC values for Guava extract stating concentration ranges of 400 to 1600 μg/mL and 1250 to 3000 μg/mL [71, 75], respectively. However, little is still known about the potential of plant extracts against *Vibrio* species. In fact, there is a dearth of literature related to the cellular viability results (CFU/mL) in biofilm inhibition and eradication assays. This becomes a challenge when comparing with the existing literature and demonstrates the importance of the present study, where the reduction in the cellular viability of *Vibrio* biofilms is equal to or even greater than tetracycline and ceftriaxone antibiotics even though the extracts are diluted in an aqueous solvent and their compounds were not purified. Future studies should evaluate the antibiofilm activity of the identified compounds of the present study.

Moreover, Guava and Eucalyptus extracts contain a variety of compounds already reported in the literature [81–83], including flavonoids, isoflavonoids, and anthocyanins. In the present study, we found cypellogin A or B and cypellocarpin C in Eucalyptus extract in agreement with the literature [58]. Likewise, our identification of guavinoside C, guavinoside A isomer, and guavinoside B in Guava extract is consistent with Rojas-Garbanzo and colleagues' study [84]. Regarding these molecules, few specific antimicrobial assays using purified compounds were found, reporting mainly antimicrobial activities by guavinoside-like compounds [85]. On the other hand, other identified molecules in the present study were widely found in other plants and already demonstrated antimicrobial properties in previous studies [86, 87]. More exactly, puerarin, quercetin, quercetin glucoside, and rutin showed specific antimicrobial activity against *Vibrio* species including *V. cholerae* [86, 88–90]. The antimicrobial activity could be attributed to various mechanisms and the presence of multiple compounds in these plant extracts could lead to synergistic effects that enhance their antibiofilm effects. The mechanisms involving flavonoids have been discussed by other authors [34, 91–93]. In general, flavonoids can suppress nucleic acid synthesis and affect the cytoplasmic membrane as well as other possible effects. Considering the identified flavonoids in the present study, quercetin, rutin, and luteolin can induce a membrane disruption and even affect biofilm formation, while quercetin and similar flavonoids can actually disrupt bacterial fatty acid metabolism [34, 92]. Similarly, anthocyanins (like cyanidin derivates found in Guava) can also affect biofilm formation probably by their chelating properties and interfere with bacterial ATP synthesis [92, 94]. For many other flavonoids or polyphenols, no specific data is known regarding effects or mechanisms, but it is safe to assume that will share similar effects as observed in other flavonoids. The combination of these effects could weaken bacteria and lead to their lysis and death. Moreover, considering the low resistance of bacteria to flavonoids [34, 95, 96] as well as the increment in antibiotic effects when combined with flavonoids [93, 97], it is important to explore further these and other natural product extracts to reduce the industrial use of conventional antibiotics or other chemical compounds with great impact on the environment. Therefore, it is important to carry out other studies to fully understand the precise mechanisms behind the antimicrobial activity of these extracts and their ability to eliminate *Vibrio*-associated biofilms.

## Limitations and further recommendations

In the present work, we indicate the biological activity of the Eucalyptus (*Eucalyptus globulus*) and Guava (*Psidium guajava*) leaf extracts. However, we don't know which of the identified compounds is responsible for the activity. Further studies need to be done to fraction the extracts with further activity evaluation and/or the direct biological activity evaluation of the pure identified compounds. Moreover, future experiments could be designed to determine the mechanism behind biological activity. Another shortcoming of the present study was the lack of *in vitro* tests for toxicological evaluation on hemolytic shrimp cells and *in vivo* assessments within the shrimp industry in Ecuador to verify the effectiveness and define optimal treatment ranges against *Vibrio*-associated biofilms to eliminate their continuous persistence in aquaculture. In addition, the CFU assays allowed to evaluate viable cells, but it cannot differentiate between live but non-culturable cells and dead cells. Therefore, CFU alone is not enough to assess biofilm inhibition and eradication, but it must be combined with other methodologies, such as biomass and LIVE/DEAD assays. Likewise, future research should explore the practical applications of these identified compounds, more exactly cypellogin A, cypellogin B, and cypellocarpin C from Eucalyptus extract, as well as guavinoside A, B, and C compounds from Guava extract. Several studies tried to develop alternative treatments to fight vibriosis and

antibiotic resistance [98]. However, there is still a need to focus on their practical application. The use of natural compounds from plant extracts is one of the most promising alternatives for the treatment and prevention of diseases in aquaculture. These compounds are eco-friendly allowing them to support the economy and reducing food spoilage. Overall, our results from aqueous extracts of Guava and Eucalyptus plants suggested a viable alternative in the ongoing quest for new alternative treatments against vibriosis in the shrimp industry serving as natural antimicrobial agents, cost-effective, sustainable, and environmentally friendly approach in Ecuador.

## Supporting information

**S1 Fig. Descriptive chromatograms obtained in Eucalyptus (*Eucalyptus globulus* Labill.) and Guava (*Psidium guajava* L.) extracts during HPLC-DAD-MS analysis.**
(DOCX)

**S2 Fig. Fluorescence microscopy of the best conditions in biofilm formation by *V. parahaemolyticus* and *V. cholerae*.** Biofilms of *Vibrio parahaemolyticus* (VP) illustrated in the best growth conditions, (at (24°C), (24h), and initial inoculum of 0.5 McFarland) by fluorescence microscopy using Live/Dead Invitrogen staining. The original image was zoomed at 1:10 to observe the cells in the biofilm to compare the total live and dead cells. An Olympus BX50 microscope with 100X magnification was used, images were obtained with AmScope software, and images were merged with Fiji-ImageJ software.
(DOCX)

**S1 Table. Parameters created by desirability functions.**
(DOCX)

**S2 Table. Optimization of the growth conditions of the *Vibrio* species.** The Minitab statistical program calculates an optimal solution that serves as the starting point for the plot, and the settings can be modified interactively to determine how different settings affect *Vibrio* biofilm growth responses using composite desirability is the weighted geometric mean of the individual desirability for the responses. Minitab determines optimal settings for the variables by maximizing the composite desirability. For biofilm formation, we sought to maximize the response of the variables with higher composite desirability.
(DOCX)

**S3 Table. Summary of the mean and median results of the present study from *Vibrio parahaemolyticus* and *Vibrio cholerae* biomass, cell viability, and total cells count assays.** Evaluation of the *in vitro* biofilm formation of two *Vibrio* species (*Vibrio parahaemolyticus* and *Vibrio cholerae*). At least six assays with quintuplicate biofilms samples were performed on different days. For the evaluation of the data, normality, and data transformation tests were performed in order to obtain a parametric analysis of all the data with the Minitab program. The mean, standard deviation, and minimum and maximum range of the trials are shown in the table. All OD measurements by PBS suspension and CV staining were adjusted by subtracting the absorbance measurements of sterility controls from the absorbance measurements of biofilm samples.
(DOCX)

**S4 Table. Summary of results of reducing model of ANOVA statistical analysis of biomass, viability, total cells counting assays in *Vibrio parahaemolyticus* and *Vibrio cholerae*.** Variables, 1: Species; 2: Temperature; 3: Time; and 4: Initial Inoculum. A normalization test and data transformation were performed to obtain normalized data for a parametric test of biofilm

growth. A multivariate ANOVA statistical analysis was performed to evaluate biofilm formation in TSB plus 1% NaCl differences between statistical values of multiple variables. The degrees of freedom (DF) indicate the number of observations free to vary, the adjusted sum of squares (SC) indicates the total sum of the variation or deviation contributed by the variables, the adjusted mean square (MC) indicates the variation that exists between the variables, the F value indicates the significance that a variable contributes to the system and the relationship between the variables (the higher the F value the greater the significance), and the *p* value equal to or less than 0.05 indicates that the null hypothesis is false and variables are significant for biofilm growth.
(DOCX)

**S5 Table. Summary of the statistical analysis and intervariable correlations by Tukey and Fisher methods in biofilm formation of *V. parahaemolyticus* (VP) and *V. cholerae* (VC).** The Tukey and Fisher methods of intervariable comparison create confidence intervals for all pairwise differences in the means of the levels of the variables. A *p* value less than or equal to 0.05 indicates whether there is a significant difference between the pairs of variable levels.
(DOCX)

**S6 Table. Summary table of Minimum Inhibition Concentration (MIC).**
(DOCX)

**S7 Table. Summary table of the biofilms inhibitory and eradication concentration.** Summary table with the absorbance values at 630 nm, CFU, percentage of inhibition and eradication, and their respective standard deviation values obtained in the Biofilm tests carried out on 2 species of *Vibrio* (*Vibrio parahaemolyticus* and *Vibrio cholerae*), carried out with 2 commercial antibiotics from different chemical families (1. Tetracycline, 2. Ceftriaxone) and 2 plant extracts (Eucalyptus and Guava).
(DOCX)

## Acknowledgments

Special recognition deserves all colleagues of the Microbiology Institute of Universidad San Francisco de Quito (IM-USFQ), COCIBA, and the Research Office of USFQ for their support in this study. We are grateful to David Valencia and Darío Cueva for their assistance in fluorescence microscopy assays.

## Author Contributions

**Conceptualization:** Lourdes Orejuela-Escobar, Eduardo Tejera, António Machado.

**Data curation:** Nicolás Renato Jara-Medina, Diego F. Cisneros-Heredia, Rebeca Cortez-Zambrano, Nelson Miranda-Moyano, António Machado.

**Formal analysis:** Nicolás Renato Jara-Medina, Dario Fernando Cueva, Ariana Cecibel Cedeño-Pinargote, Arleth Gualle, Daniel Aguilera-Pesantes, Diego F. Cisneros-Heredia, Rebeca Cortez-Zambrano, Nelson Miranda-Moyano, Eduardo Tejera, António Machado.

**Funding acquisition:** António Machado.

**Investigation:** Nicolás Renato Jara-Medina, Dario Fernando Cueva, Ariana Cecibel Cedeño-Pinargote, Arleth Gualle, Daniel Aguilera-Pesantes, Miguel Ángel Méndez, Lourdes Orejuela-Escobar, António Machado.

**Methodology:** António Machado.

**Project administration:** António Machado.

**Resources:** António Machado.

**Software:** Eduardo Tejera, António Machado.

**Supervision:** Eduardo Tejera, António Machado.

**Validation:** Eduardo Tejera, António Machado.

**Visualization:** Eduardo Tejera, António Machado.

**Writing – original draft:** Nicolás Renato Jara-Medina, Ariana Cecibel Cedeño-Pinargote, Lourdes Orejuela-Escobar, Eduardo Tejera, António Machado.

**Writing – review & editing:** Miguel Ángel Méndez, Lourdes Orejuela-Escobar, Eduardo Tejera, António Machado.

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
