## [Decision Letter · Decision Letter 0]

23 Feb 2024

PONE-D-24-02362Eco-alternative treatments for Vibrio parahaemolyticus and V. cholerae biofilms from shrimp industry through Eucalyptus and Guava extracts: A road for an Ecuadorian sustainable economyPLOS ONE

Dear Dr. Machado,

Thank you for submitting your manuscript to PLOS ONE. After careful consideration, we feel that it has merit but does not fully meet PLOS ONE’s publication criteria as it currently stands. Therefore, we invite you to submit a revised version of the manuscript that addresses the points raised during the review process.

We look forward to receiving your revised manuscript.

Kind regards,

Arumugam Sundaramanickam, PhD

Academic Editor

PLOS ONE

2. Please keep your tables as part of your main manuscript and remove the individual files. Please note that supplementary tables (should remain/ be uploaded) as separate "supporting information" files

Reviewers' comments:

Reviewer's Responses to Questions

**Comments to the Author**

1. Is the manuscript technically sound, and do the data support the conclusions?

Reviewer #1: Partly

Reviewer #2: Yes

Reviewer #3: Partly

2. Has the statistical analysis been performed appropriately and rigorously? 

Reviewer #1: Yes

Reviewer #2: Yes

Reviewer #3: I Don't Know

3. Have the authors made all data underlying the findings in their manuscript fully available?

Reviewer #1: Yes

Reviewer #2: Yes

Reviewer #3: Yes

4. Is the manuscript presented in an intelligible fashion and written in standard English?

Reviewer #1: Yes

Reviewer #2: Yes

Reviewer #3: No

5. Review Comments to the Author

Reviewer #1: Review comments for Plos One manuscript no. PONE-D-24-02362 “Eco-alternative treatments for Vibrio parahaemolyticus and V. cholerae biofilms from shrimp industry through Eucalyptus and Guava extracts: A road for an Ecuadorian sustainable economy.”

This paper evaluated the impact of temperature, time, and initial inoculum on the biofilm development of two Vibrio species, V. parahaemolyticus and V. cholerae, using a multifactorial experimental design. It also reports the effects of two aqueous extracts, Eucalyptus and Guava aqueous leaf extracts, as well as tetracycline and ceftriaxone, on bacterial biofilm formation and eradication. The paper is well written and the work, in general, is OK. However, there are some aspects that are not clear and need to be addressed before it is suitable for publication in PLOS ONE. Please see those comments, questions, and suggestions in the main manuscript file.

Reviewer #2: The manuscript presents meticulously designed experiments with a logical structure, promising to capture the interest of the scientific community. The findings contribute significantly to the field.

However, the reviewer highlighted areas for improvement. Firstly, the introduction should clearly articulate the main objective and underscore the novelty of the research. A comprehensive introduction will aid readers in understanding the study's significance.

Secondly, there is a need for improvement in figure quality to enhance clarity and readability. High-quality figures are essential for facilitating comprehension, and efforts should be directed toward refining the visual representations.

Lastly, the discussion section requires attention. Some parts are challenging for readers to comprehend, so efforts should be made to enhance coherence and clarity, ensuring a better understanding for the audience.

Specific comments:

Comment 1: In line 171, 202, minutes~min. Please check throughout the manuscript.

Comment 2: In line 246-251, Please check, rewrite.

Comment 3: In line 466-470, This are not correct, please check.

Reviewer #3: This MS is well written but lacks many scientific aspects. Scientifically the major issue on this MS is it was not followed up through bioassay guided purification. The compounds are identified by MS based on the major abundance is inappropriate since it brings lot of questions. What compound is responsible for the bioactivity. Authors did not test any of the predicted compounds leads to novelty, since previously reports are available on these two plants with against Vibrio parahaemolyticus.

Guava (Psidium guajava) leaf extract enhances immunity, growth, and resistance against Vibrio parahaemolyticus in white shrimp Penaeus vannamei - PubMed (nih.gov)

Antibacterial activity of guava (Psidium guajava L.) and Neem (Azadirachta indica A. Juss.) extracts against foodborne pathogens and spoilage bacteria - PubMed (nih.gov)

Microorganisms | Free Full-Text | Inhibitory Activity of Essential Oils against Vibrio campbellii and Vibrio parahaemolyticus (mdpi.com)

Anti-vibrio and immune-enhancing activity of medicinal plants in shrimp: A comprehensive review - ScienceDirect

Effectiveness of medicinal plant extracts against Vibrio spp. in shrimp aquaculture - Ghosh - 2021 - Aquaculture Research - Wiley Online Library

While w.r.t. V. cholerae so many literatures available even with antibiofilm properties.

Other comments:

1. here are several grammatical errors throughout the manuscript. I recommend having a native English speaker review the manuscript for clarity and correctness.

2. This MS lack the detail information on indigenous uses in aquaculture or why these plants were chosen. Providing this context could strengthen your argument and make the study more relatable to readers.

3. The taxonomic identification of the plants is missing, and no voucher specimen appears to have been deposited. This is essential for the reproducibility and verification of your study. Why the plant is not identified to species level?

4. What are the yield value of the extract.

5. Line nos : 169-186, deals with biofilm but the cited reference is for Candida sp. This raises questions about the authenticity of the protocols followed. Please ensure that the references cited are relevant to your study.

21. “Atiencia-Carrera MB, Cabezas-Mera FS, Vizuete K, Debut A, Tejera E, Machado A. Evaluation 788 of the biofilm life cycle between Candida albicans and Candida tropicalis. Front Cell Infect 789 Microbiol. 2022;12: 953168. doi:10.3389/fcimb.2022.953168” and 24. “Lohse MB, Gulati M, Johnson AD, Nobile CJ. Development and regulation of single-and multi 798 species Candida albicans biofilms. Nat Rev Microbiol. 2018;16: 19–31. 799 doi:10.1038/NRMICRO.2017.107”

“2.4.1 Crystal violet staining” is well known but lacks here with proper citation.

6. Line no. 154 “0.5 mL of 0.1 % formic acid/acetonitrile (70:30, v/v), does it mean 30% water? On line 154, you mention a 0.1% formic acid/acetonitrile (70:30, v/v) solution. If this implies 30% water, a water extract would be expected to dissolve in 1-5% acetonitrile rather than 70% acetonitrile. Please provide details of the HPLC LC protocols and chromatogram as supplementary information.

7. The figures in the manuscript are of low quality and difficult to read. Please consider improving the resolution for better readability.

8. The plant name is not italics in many cases, and authors citation is italics. Many time it is abbreviated and again full name is mentioned.

9. References are not uniform format. Many biological names are not italics.

6. PLOS authors have the option to publish the peer review history of their article (what does this mean?). If published, this will include your full peer review and any attached files.

Reviewer #1: No

Reviewer #2: No

Reviewer #3: No

---

## [Author Response · Author response to Decision Letter 0]

11 Apr 2024

PONE-D-24-02362

Eco-alternative treatments for Vibrio parahaemolyticus and V. cholerae biofilms from shrimp industry through Eucalyptus (Eucalyptus globulus) and Guava (Psidium guajava) extracts: A road for an Ecuadorian sustainable economy

Dear Editor and Reviewers,

Thank you very much for the comments regarding our original version of the submitted manuscript.

We have taken all reviewers’ comments and suggestions into careful consideration and revised the manuscript accordingly. On the following pages, please find our point-to-point responses to the editors’ and reviewers’ concerns in the order that they were originally listed, and details of the lines and pages on which the changes have been made.

It is important to mention that several rectifications were made according to reviewer concerns and comments. The major corrections were realized by adding the voucher specimens of Guava (Psidium guajava L.) and Eucalyptus (Eucalyptus globulus Labill.) leaf samples used in this study under the codes #35004 and #35005, respectively, and the authorization for collection of biological specimens (MAATE-ARSFC-2024-0328) issued by the Ministry of Environment, Water and Ecological Transition of Ecuador MAATE (see the new subsection 2.3 Ethics statement of the revised manuscript). This amendment was realized together with the Herbario de Botánica Económica del Ecuador of the USFQ; therefore, three coauthors were added to the revised manuscript due to the work involved in the present study. In addition, we improved the description of several procedures in the Materials and Methods section, and the resolution of figure pictures was also improved and corrected in the Results section through PLOS ONE guidelines and the Pace tool (https://pacev2.apexcovantage.com/), respectively. New information was added in the Introduction and Discussion sections about the Ecuadorian context and the Public Health threats involving Vibrio parahaemolyticus and V. cholerae, as well as the low-cost and available amount of Guava (Psidium guajava) and Eucalyptus (Eucalyptus globulus) plants as natural antimicrobial agents, cost-effective, sustainable, and environmentally friendly approach in Ecuador. Likewise, we added a new supplementary figure S1 to illustrate the descriptive HPLC chromatograms and the yield value of the plant extracts in the Results section. Several studies were added in the Introduction and Discussion sections to clarify the main goal and the novelty of the present study. Finally, we also realized intensive English editing together with English native speakers of our institution to improve the manuscript, checking the uniform format of all references and putting all biological names in the italic form.

We hope that the revised manuscript will be considered suitable for PLOS ONE’s publication criteria If there are further questions, please let us know. Thank you very much.

Sincerely,

António Machado (corresponding author)

Corrected Manuscript

Detailed Response to Editor and Reviewers

We want to thank the Editor and all reviewers for their constructive comments and thoughtful suggestions that allowed us to improve the original manuscript. Based on their comments and suggestions, we have made several modifications and corrections as described in the report below:

Reviewer 1

Comments to the Authors

Reviewer #1: Review comments for Plos One manuscript no. PONE-D-24-02362 “Eco-alternative treatments for Vibrio parahaemolyticus and V. cholerae biofilms from shrimp industry through Eucalyptus and Guava extracts: A road for an Ecuadorian sustainable economy.”

This paper evaluated the impact of temperature, time, and initial inoculum on the biofilm development of two Vibrio species, V. parahaemolyticus and V. cholerae, using a multifactorial experimental design. It also reports the effects of two aqueous extracts, Eucalyptus and Guava aqueous leaf extracts, as well as tetracycline and ceftriaxone, on bacterial biofilm formation and eradication. The paper is well written and the work, in general, is OK. However, there are some aspects that are not clear and need to be addressed before it is suitable for publication in PLOS ONE. Please see those comments, questions, and suggestions in the main manuscript file.

Authors' answers: We want to thank Reviewer 1 for his/her constructive comments and thoughtful suggestions that allowed us to improve the original manuscript. All unclear aspects were addressed in the revised manuscript. We are pleased to inform you that thanks to your recommendations we have made the following changes:

Lines 40-42- What is/are the rationale (s) for choosing these two species? V. parahaemolyticus form biofilm at a relatively high cell density, while V. cholerae form biofilm at low cell density. The genetic controls of their biofilm formations are different.

Authors' answers: As well-appointed by Reviewer 1, the genetic controls of V. parahaemolyticus and V. cholerae biofilm formations are different, and detailed information was added in the Introduction section as well-suggested in a further comment (please check lines 104-113 on page 3 of the revised manuscript). Meanwhile, the rationale for choosing these two species is the endemic prevalence of these Vibrio species in the shrimp farms in Southern Coastal Ecuador, as previously reported by Ryan et al. (2018) and Sperling et al. (2015), and the collaboration partner Clydent SA suggestion to selecting these species isolated from shrimp farms. This information was added in the Abstract and Introduction section to clarify the non-familiar Readers of the Ecuadorian context. Thank you. Please check the amendments in lines 43-44 in the Abstract section and lines 121-123 on page 4 in the Introduction section of the revised manuscript.

References

Ryan SJ, Stewart-Ibarra AM, Ordóñez-Enireb E, Chu W, Finkelstein JL, King CA, Escobar LE, Lupone C, Heras F, Tauzer E, Waggoner E, James TG, Cárdenas WB, Polhemus M. Spatiotemporal Variation in Environmental Vibrio cholerae in an Estuary in Southern Coastal Ecuador. Int J Environ Res Public Health. 2018 Mar 10;15(3):486. doi: 10.3390/ijerph15030486.

Sperling L, Alter T, Huehn S. Prevalence and Antimicrobial Resistance of Vibrio spp. in Retail and Farm Shrimps in Ecuador. J Food Prot. 2015 Nov;78(11):2089-92. doi: 10.4315/0362-028X.JFP-15-160.

Lines 42-44- Why use CFU, not their biofilm biomasses, instead?

Authors' answers: As well-asked by Reviewer 1, the CFU counting assays were chosen as a reference because it is considered a more accurate quantitative methodology in the literature (such as Wilson et al., 2017 or Haas et al., 2023) and this approach was already realized in a previous study with Candida-related biofilms (Atiencia et al., 2022). However, CFU quantification allows to evaluate viable cells, but it cannot differentiate between live but non-culturable cells and dead cells. Therefore, CFU alone is not enough to assess biofilm inhibition and eradication, but it must be combined with other methodologies, such as biomass and LIVE/DEAD assays. The clarification of this limitation was added in lines 776-779 on page 19 of the Discussion section. 

References

Atiencia-Carrera MB, Cabezas-Mera FS, Vizuete K, Debut A, Tejera E, Machado A. Evaluation of the biofilm life cycle between Candida albicans and Candida tropicalis. Front Cell Infect Microbiol. 2022 Aug 18;12:953168. doi: 10.3389/fcimb.2022.953168.

Wilson C, Lukowicz R, Merchant S, Valquier-Flynn H, Caballero J, Sandoval J, Okuom M, Huber C, Brooks TD, Wilson E, Clement B, Wentworth CD, Holmes AE. Quantitative and Qualitative Assessment Methods for Biofilm Growth: A Mini-review. Res Rev J Eng Technol. 2017 Dec;6(4): http://www.rroij.com/open-access/quantitative-and-qualitative-assessment-methods-for-biofilm-growth-a-minireview-.pdf.

Haas B, James S, Parker AE, Gagnon MC, Goulet N, Labrie P. Comparison of quantification methods for an endoscope lumen biofilm model. Biofilm. 2023 Oct 20;6:100163. doi: 10.1016/j.bioflm.2023.100163.

Line 54- plant extracts?

Authors' answers: As well-appointed by Reviewer 1, we replaced “extract plants” with “plant extracts”. Please check the amendment in lines 58-59 on page 2 of the revised manuscript. We apologize for this typo error and all text of the manuscript was carefully revised in the new version.

Lines 95-99- The genetic controls of V. parahaemolyticus and V. cholerae biofilm formations are different. The authors should provide detailed information here.

Authors' answers: We agree with Reviewer 1 recommendation and therefore we provided detailed information about the differences in genetic controls of V. parahaemolyticus and V. cholerae biofilm formations. Please check the new information in lines 104-113 on page 3 of the revised manuscript, more exactly:

“Although all Vibrio species are ubiquitous in aquatic ecosystems and can establish strong biofilms, Vibrio species can also differ in the mechanisms and regulation of their biofilm formation [19]. For example, V. cholerae biofilms are positively controlled by VpsR and VpsT regulators via the transcriptional control of vps genes, where VpsR is greater than VpsT. Meanwhile, V. parahaemolyticus biofilms are positively regulated via expression of cps genes, which are negatively regulated by a homolog of VpsT known as CpsS. In the absence of CpsS and CpsR (VpsR homolog), the positive cps gene expression leads to the V. parahaemolyticus biofilm formation. Thus, V. cholerae and V. parahaemolyticus use similar proteins, but they function in the opposite sense and to different degrees: CpsS is the dominant negative regulator in V. parahaemolyticus whereas VpsT is a positive co-regulator in V. cholerae [19,20].”

References

19. Yildiz FH, Visick KL. Vibrio biofilms: so much the same yet so different. Trends in Microbiology. 2009. pp. 109–118. doi:10.1016/j.tim.2008.12.004 https://pubmed.ncbi.nlm.nih.gov/19231189/

20. Teschler JK, Zamorano-Sánchez D, Utada AS, Warner CJA, Wong GCL, Linington RG, et al. Living in the matrix: Assembly and control of Vibrio cholerae biofilms. Nature Reviews Microbiology. Nature Publishing Group; 2015. pp. 255–268. doi:10.1038/nrmicro3433 https://pubmed.ncbi.nlm.nih.gov/25895940/

Lines 100-102- Is this sentence correct? And please also cite the original work. 

Authors' answers: As well-suggested by Reviewer 1, we checked the data of the original sentence and we properly cited the original works, more exactly, Tran et al. (2013) and De Schryver et al. (2014). Please check the amendment in lines 114-117 on page 3 of the revised manuscript, more exactly:

Original sentence

“The main diseases that affect shrimp are early mortality shrimp-acute hepatopancreatic necrosis disease (EMS – AHPND) and causes 100% of mortality, while vibriosis causes 70% of mortality but damages the shrimp´s oral cavity and appendages [19].” 

Original reference

Previous reference 19. Sajali USBA, Atkinson NL, Desbois AP, Little DC, Murray FJ, Shinn AP. Prophylactic properties of biofloc- or Nile tilapia-conditioned water against Vibrio parahaemolyticus infection of whiteleg shrimp (Penaeus vannamei). Aquaculture. 2019;498: 496–502. doi:10.1016/j.aquaculture.2018.09.002

Revised sentence

“The main diseases that affect shrimp are early mortality shrimp-acute hepatopancreatic necrosis disease (EMS – AHPND) affects shrimp postlarvae within 20–30 days after stocking and frequently causes up to 100% mortality [21,22], while the mortality of vibriosis can range between 70-100% but damages the shrimp´s oral cavity and appendages [5,23].”

Updated and rectified references 

21. De Schryver P, Defoirdt T, Sorgeloos P. Early Mortality Syndrome Outbreaks: A Microbial Management Issue in Shrimp Farming? PLoS Pathog. 2014;10. doi:10.1371/journal.ppat.1003919 https://doi.org/10.1371/journal.ppat.1003919

22. Tran L, Nunan L, Redman RM, Mohney LL, Pantoja CR, Fitzsimmons K, et al. Determination of the infectious nature of the agent of acute hepatopancreatic necrosis syndrome affecting penaeid shrimp. Dis Aquat Organ. 2013;105: 45–55. doi:10.3354/dao02621 https://pubmed.ncbi.nlm.nih.gov/23836769/

23. De La Peña LD, Cabillon NAR, Catedral DD, Amar EC, Usero RC, Monotilla WD, et al. Acute hepatopancreatic necrosis disease (AHPND) outbreaks in Penaeus vannamei and P. monodon cultured in the Philippines. Dis Aquat Organ. 2015;116: 251–254. doi:10.3354/dao02919 https://pubmed.ncbi.nlm.nih.gov/26503780/

5. Sanches-Fernandes GMM, Sá-Correia I, Costa R. Vibriosis Outbreaks in Aquaculture: Addressing Environmental and Public Health Concerns and Preventive Therapies Using Gilthead Seabream Farming as a Model System. Front Microbiol. 2022;13. doi:10.3389/fmicb.2022.904815

Lines 102-104- Please also check this sentence for accuracy, and the paper cited. It is well known that V. harveyi and V. parahaemolyticus are the two most important Vibrio species that cause disease in shrimp.

Authors' answers: As well-appointed by Reviewer 1, we examined the accuracy of the original sentence and we rectified and properly cited the original works, more exactly, Ryan et al. (2018) and Sperling et al. (2015). We rectified the main shrimp-related pathogens and the human-related infections, also explaining the Ecuadorian context and justifying the selection of V. parahaemolyticus and V. cholerae in the present study. We apologize for the mistake and confusing sentence. Please check the amendment in lines 117-123 on pages 3-4 of the revised manuscript, more exactly: 

Original sentence

“These diseases are caused by pathogenic Vibrio species, being Vibrio alginolyticus, V. parahaemolyticus, V. cholerae, and V. vulnificus the main pathogens isolated [16].” 

Original reference

16. Flemming H-C, van Hullebusch ED, Neu TR, Nielsen PH, Seviour T, Stoodley P, et al. The biofilm matrix: multitasking in a shared space. Nature Reviews Microbiology 2022. 2022; 1–775 17. doi:10.1038/s41579-022-00791-0

Revised sentence

“Therefore, several Vibrio species can significantly reduce shrimp production where V. parahaemolyticus, V. harveyi, V. alginolyticus, and V. vulnificus are commonly isolated in these shrimp farms [24]. In addition, certain Vibrio species can emerge as a serious threat to human health, such as V. parahaemolyticus, V. cholerae, and V. alginolyticus [25]. In Ecuador, the prevalence of V. parahaemolyticus and V. cholerae in shrimps at retail, farms, and estuaries are alarming and represent an eminent and serious public health risk [26,27].”

Updated and rectified references 

24. de Souza Valente C, Wan AHL. Vibrio and major commercially important vibriosis diseases in decapod crustaceans. J Invertebr Pathol. 2021;181. doi:10.1016/j.jip.2020.107527

25. Robert-Pillot A, Copin S, Himber C, Gay M, Quilici ML. Occurrence of the three major Vibrio species pathogenic for human in seafood products consumed in France using real-time PCR. Int J Food Microbiol. 2014;189: 75–81. doi:10.1016/j.ijfoodmicro.2014.07.014

26. Ryan SJ, Stewart-Ibarra AM, Ordóñez-Enireb E, Chu W, Finkelstein JL, King CA, et al. Spatiotemporal variation in environmental Vibrio cholerae in an estuary in southern coastal Ecuador. Int J Environ Res Public Health. 2018;15. doi:10.3390/ijerph15030486. https://pubmed.ncbi.nlm.nih.gov/29534431/

27. Sperling L, Alter T, Huehn S. Prevalence and antimicrobial resistance of Vibrio spp. in retail and farm shrimps in Ecuador. J Food Prot. 2015;78: 2089–2092. doi:10.4315/0362-028X.JFP-15-160. https://pubmed.ncbi.nlm.nih.gov/26555534/

Lines 112-113- “Some alternative strategies to control Vibrio species and their biofilm formation is using probiotics and natural products in mature ecosystems [2,6].” Please check this "Yatip, P., Teja, D. Nitin Chandra, Flegel, T. W. and Soowannayan, C. (2018) Extract from the fermented soybean product Natto inhibits Vibrio biofilm formation and reduces shrimp mortality from Vibrio harveyi infection, Fish & Shellfish Immunology 72:348-355" and other papers from the same group.

Authors' answers: As well-recommended by Reviewer 1, we added some studies on alternative strategies to control Vibrio-related biofilms or infections, such as Yatip et al. (2018), Karnjana et al. (2019), and Kamble et al. (2022). Thank you for your recommendations that allow us to improve the original manuscript. Pleas

---

## [Decision Letter · Decision Letter 1]

7 May 2024

Eco-alternative treatments for Vibrio parahaemolyticus and V. cholerae biofilms from shrimp industry through Eucalyptus (Eucalyptus globulus) and Guava (Psidium guajava) extracts: A road for an Ecuadorian sustainable economy

PONE-D-24-02362R1

Dear Dr. Machado,

We’re pleased to inform you that your manuscript has been judged scientifically suitable for publication and will be formally accepted for publication once it meets all outstanding technical requirements.

Kind regards,

Arumugam Sundaramanickam, PhD

Academic Editor

PLOS ONE

Additional Editor Comments (optional):

Reviewers' comments:

Reviewer's Responses to Questions

**Comments to the Author**

1. If the authors have adequately addressed your comments raised in a previous round of review and you feel that this manuscript is now acceptable for publication, you may indicate that here to bypass the “Comments to the Author” section, enter your conflict of interest statement in the “Confidential to Editor” section, and submit your "Accept" recommendation.

Reviewer #1: All comments have been addressed

Reviewer #2: All comments have been addressed

2. Is the manuscript technically sound, and do the data support the conclusions?

Reviewer #1: Yes

Reviewer #2: Yes

3. Has the statistical analysis been performed appropriately and rigorously? 

Reviewer #1: Yes

Reviewer #2: Yes

4. Have the authors made all data underlying the findings in their manuscript fully available?

Reviewer #1: Yes

Reviewer #2: Yes

5. Is the manuscript presented in an intelligible fashion and written in standard English?

Reviewer #1: Yes

Reviewer #2: Yes

6. Review Comments to the Author

Reviewer #1: The authors have satisfactorily responded to all my comments and suggestions. Therefore, the paper should be acceptable for publication in PlosOne.

Reviewer #2: The manuscript is well written, and they submitted their updated manuscript according to my suggestions.

7. PLOS authors have the option to publish the peer review history of their article (what does this mean?). If published, this will include your full peer review and any attached files.

Reviewer #1: No

Reviewer #2: No
